# Gestalt Reasoning Machines: Structured Perception for Neuro-Symbolic Inference

## Abstract

This paper introduces Gestalt Reasoning Machines (GRMs), a novel neuro-symbolic framework that integrates Gestalt principles to enhance reasoning models with perception capabilities similar to human cognition. Traditional models, which rely on large datasets and complex computations, often overlook the crucial human cognitive function of grouping, resulting in inefficiencies when dealing with abstract concepts. GRMs address this challenge by incorporating a grouping mechanism grounded in Gestalt principles, enabling the system to recognize and reason over complex visual patterns that are otherwise difficult to capture through object-level features alone. This grouping capability allows GRMs to identify higher-order structures and relational configurations that are essential for human-like reasoning. We demonstrate that GRMs outperform purely neural baselines by leveraging logic-based reasoning infused with perceptual grouping cues, offering a more interpretable and cognitively aligned approach. Our contributions include the design of GRMs and the empirical validation of their effectiveness in visual reasoning tasks that demand structured perception.

## 1 Introduction

Human visual perception excels at organizing complex scenes into meaningful structures through perceptual grouping. *Gestalt principles*—such as proximity, similarity, and continuity—explain how individuals organize visual elements into coherent wholes rather than processing them as isolated components. These principles, rooted in psychology research (Koffka, 1935; Wertheimer, 1938; Palmer, 1999; Ellis, 1999), are fundamental to how humans efficiently parse and reason about visual scenes. As illustrated in Fig. 1 (right), humans naturally perceive objects based on their spatial arrangements and shared attributes, identifying patterns that enable structured understanding of complex visual environments.

Current approaches to visual reasoning typically rely on scaling up neural models with massive datasets and parameters (Kojima et al., 2022; Huang & Chang, 2023; Cheng et al., 2024; Zhang et al., 2025). However, these data-driven models struggle with abstract reasoning (Huang et al., 2024), particularly in Vision-Language Models (VLMs) where reasoning capabilities remain severely limited (Chen et al., 2025; Wüst et al., 2025). Their reasoning often lacks grounding and becomes inconsistent when processing complex multi-object scenes (Fu et al., 2024; Majumdar et al., 2024; Zhang et al., 2024a), highlighting limitations in their ability to capture structured relationships.

Neuro-symbolic approaches offer a promising alternative by integrating symbolic reasoning with neural perception. These models have demonstrated strong performance on complex reasoning tasks requiring relational inference and numerical computation over visual inputs (Yi et al., 2018; Amizadeh et al., 2020; Manhaeve et al., 2021; Marra et al., 2024). Differentiable rule learners (Evans & Grefenstette, 2018; Shindo et al., 2023; 2024b) have been successfully applied to visual reasoning challenges, discovering explicit rules to explain visual patterns through program induction (Shindo et al., 2024a; Sudhakaran et al., 2025). However, existing neuro-symbolic approaches face a critical scalability bottleneck: they lack effective *grouping* mechanisms that are fundamental to human visual perception (Han et al., 2002; Xu & Chun, 2007; Thórisson, 2019). When processing complex scenes, humans naturally organize objects according to perceptual cues like similarity and proximity, enabling efficient reasoning by reducing redundant relational computations. In contrast, most neuro-symbolic models rely on exhaustive pairwise relation generation, enumerating all possible

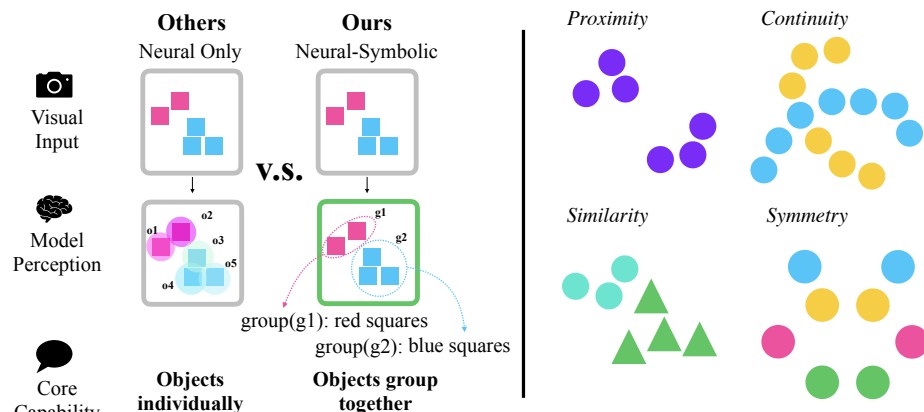

Figure 1: **Scene reasoning with Gestalt grouping. Left:** Comparison of model perception. Neural-only models process objects individually without structural organization, whereas GRM integrates perceptual grouping with symbolic reasoning, producing interpretable group-based representations (e.g., red vs. blue squares). **Right:** Examples of gestalt principles. Proximity, similarity, symmetry, and continuity. Each of these principles guide how objects are grouped into meaningful structures.

object relationships. This approach becomes computationally prohibitive and brittle as scene complexity increases. This raises the central research question: *How can we endow neuro-symbolic models with human-like grouping mechanisms that enable efficient and robust reasoning over complex visual scenes?*

We propose the *Gestalt Reasoning Machine (GRM)*, a neuro-symbolic framework that explicitly incorporates perceptual grouping into visual reasoning. Unlike conventional object-centric approaches, GRM first organizes scene elements into structured groups based on spatial and attributional patterns (Fig. 1, left). These grouped entities serve as the foundation for symbolic reasoning, allowing the model to operate over scenes in a hierarchically structured and interpretable manner. By treating scenes as compositions of perceptual groups rather than collections of isolated objects, GRM captures higher-level regularities and solves complex visual problems that traditional approaches often fail to address. Given a complex scene with multiple objects, GRMs perform rule learning to identify and group objects that share underlying patterns, simultaneously producing structured perceptual representations and symbolic programs for reasoning. The resulting structured perception is then fed to the reasoning module, which efficiently infers solutions by operating over these coherent groups rather than individual objects.

Our work makes the following key contributions:

- We introduce Gestalt Reasoning Machines (GRMs)[1], the first neuro-symbolic framework that integrates perceptual grouping with symbolic rule learning, offering a robust and interpretable approach to complex visual scene understanding.

- We develop a scalable grouping mechanism that operates directly on visual input and seamlessly integrates with symbolic reasoning, enabling GRMs to scale effectively with increasing numbers of objects by reducing unnecessary relational complexity.

- We demonstrate that GRMs significantly outperform existing neural and neuro-symbolic models on structured visual reasoning tasks, with performance gains that increase as scene complexity grows. Our results show that GRMs successfully bridge the gap between data-driven modeling and efficient structured perception.

The remainder of this paper presents related work, details the GRM architecture and training procedure, provides a comprehensive experimental evaluation, and concludes with implications for future research in neuro-symbolic visual reasoning.

---

[1]We make our code publicly available at `https://anonymous.4open.science/r/nesy_causal_p-7487`

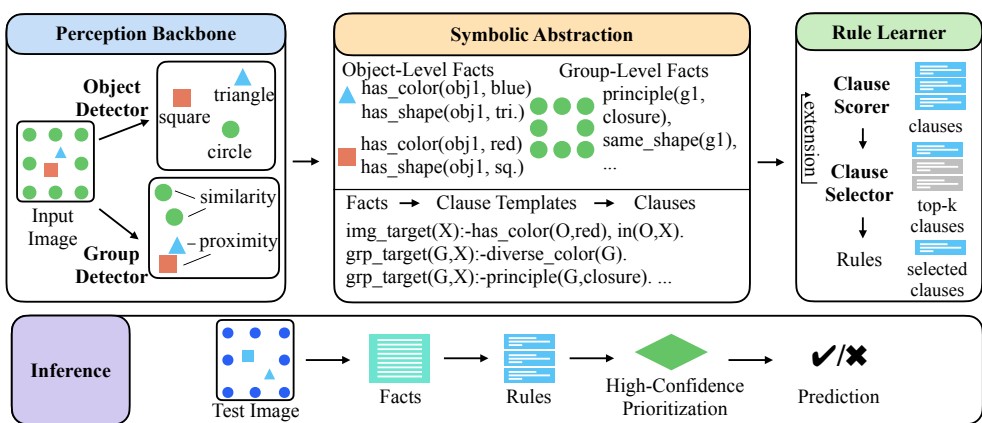

Figure 2: **Overview of the Gestalt Reasoning Machine (GRM) Pipeline.** An input image is first processed by a **Perception Backbone** that detects objects and identifies Gestalt-based group structures. Next, the **Symbolic Abstraction** module converts these perceptual features into a logical fact base and generates candidate clauses. The **Rule Learner** then performs a search over this symbolic space to discover a final set of *interpretable rules*. Finally, the **Inference** engine applies these rules to a new visual input, employing a *high-confidence prioritization strategy*. This ensures that predictions are based on highly certain, transparent rules whenever possible, falling back to a weighted aggregation of evidence from all relevant rules in more ambiguous cases.

## 2 RELATED WORK

Visual reasoning is a fundamental problem in machine learning research, leading to the development of various benchmarks (Antol et al., 2015; Johnson et al., 2017; Yi et al., 2020) and subsequent frameworks (Yi et al., 2018; Mao et al., 2019; Amizadeh et al., 2020; Hsu et al., 2023) focused on reasoning through symbolic programs and multi-modal transformers (Tan & Bansal, 2019). These benchmarks primarily aim to answer queries expressed in natural language in conjunction with visual inputs. Our work evaluates Gestalt Reasoning Models (GRMs), particularly emphasizing their ability to perform the grouping function based on Gestalt reasoning principles. Reinforcement learning has been used to enhance the reasoning capability of large Vision-Language Models (VLMs) (Liu et al., 2025; Tan et al., 2025; Zhai et al., 2024; Li et al., 2025). However, the resulting reasoning traces are not logically grounded in terms of objects (Sarch et al., 2025). Consequently, these models struggle to perform structured perception with grouping. We aim at developing the foundaiton of performing structured perception with a neuro-symbolic approach.

Additionally, Abstract Visual Reasoning (AVR) explores the capability to apply previously acquired knowledge and techniques in completely new contexts, posing unique challenges for deep neural networks (DNNs) (Hu et al., 2021; Malkinski & Mandziuk, 2023; Camposampiero et al., 2023). AVR methods have been primarily evaluated through simple abstract puzzles like Raven's progressive matrices (Raven & Court, 1998). The Kandinsky patterns framework (Müller & Holzinger, 2021) provides a unique method for generating patterns with abstract objects, which we have expanded to address Gestalt reasoning tasks. To address these challenges, neuro-symbolic rule learning frameworks have been developed, emphasizing the learning of discrete rule structures via backpropagation (Evans & Grefenstette, 2018; Minervini et al., 2020; Shindo et al., 2023; 2024b; Zimmer et al., 2023; Sha et al., 2024). These methodologies have predominantly been tested on visual arithmetic tasks or within synthetic environments tailored for reasoning (Stammer et al., 2021). Our work on Gestalt Reasoning Models (GRMs) seeks to bridge the gap between existing neuro-symbolic paradigms and elements of human cognitive processes.

Gestalt reasoning has been extensively studied in psychology (Wertheimer, 1938; Koffka, 1935; Palmer, 1999; Ellis, 1999) and has also intersected significantly with machine learning and deep learning research (Lörincz et al.; Hua & Kunda, 2020; Kim et al., 2021; Zhang et al., 2024b), although previous efforts primarily focused on convolutional neural networks. GRMs represent the first neuro-symbolic framework that explicitly encodes the grouping function in Gestalt principles.

## 3 GESTALT REASONING MACHINES

The Gestalt Reasoning Machine (GRM) is a neuro-symbolic framework that integrates perceptual organization with logical reasoning. Inspired by human visual cognition, it processes raw images to detect objects and perceptual groups, abstracts them into object- and group-level facts, and applies a logic-based rule learner to derive interpretable decision functions (Fig. 2). By combining neural perception with symbolic abstraction and rule learning, GRM achieves flexible yet interpretable reasoning over complex visual patterns. The following subsections detail its components.

### 3.1 PERCEPTION BACKBONE

The perceptual backbone of a GRM transforms raw visual input into structured symbolic and neural representations. This process operates at two levels: *object-level perception* and *group-level perception*, each combining discrete symbolic attributes with continuous neural descriptors.

**Object-Level Perception.** Given an input image $I$, the system identifies a set of perceptual objects $\mathcal{O} = \{o_1, o_2, \ldots, o_n\}$, where each object $o_i$ corresponds to a visually coherent region with binary mask $M_i \subset \mathbb{Z}^2$ indicating its spatial extent. The GRM architecture is modular with respect to the perception backbone: we instantiate it with color-based segmentation for synthetic environments and neural object proposals for natural images. Each object maintains a dual representation combining symbolic and neural features:

$$o_i = (\phi_i^{\text{sym}}, \phi_i^{\text{neu}}).$$

The symbolic features $\phi_i^{\text{sym}}$ encode high-level attributes including shape category, color, and spatial position, extracted through dedicated neural components. The neural features $\phi_i^{\text{neu}}$ capture fine-grained geometric structure through sampled contour points organized into local patches. This patch-wise representation preserves local geometric details such as corners, curves, and edge orientations that are crucial for detecting perceptual relationships. Two objects with aligned edges or similar local curvatures can be identified through matching patches, while the full contour enables robust shape classification. This dual representation thus bridges abstract reasoning with detailed perceptual analysis, supporting both symbolic manipulation and learned grouping operations in downstream modules. More detailed backbone architecture is available in App. A.

**Group-Level Perception.** Beyond individual objects, GRMs organize visual elements into perceptual groups following Gestalt principles. We implement five principles: *proximity*, *similarity*, *closure*, *symmetry*, and *continuity*. Rather than hard-coding separate mechanisms, we employ a unified architecture with a shared encoder and lightweight principle-specific heads.

The shared encoder $f$ transforms each object $o_i$ into a fixed-length embedding $\mathbf{o}_i = f(o_i)$. To capture the global context for grouping decisions, the network evaluates pariwise relationships while condditioning on all other objects in the scene. For each principle $p$, the network estimates affinities between object pairs:

$$s_p(o_i, o_j, I) = \sigma\big(h_p(\mathbf{o}_i, \mathbf{o}_j, \mathbf{o}_{ij}^*)\big),$$

where $h_p$ is a principle-specific MLP, $\sigma$ is the sigmoid activation, and $\mathbf{o}_{ij}^*$ represents the global context computed as the mean embedding of all other objects in scene $I$ (see Fig. 3). This mean pooling is permutation-invariant and keeps the scale of $\mathbf{o}_{ij}$ independent of the number of objects, which is important for comparing scenes with different object counts. It plays the role of a simple, scene-level context that informs whether a local pair is typical or exceptional within its neighborhood (Gaifman, 1982). A natural alternative is to use a Transformer over the set of object embeddings, which can in principle handle variable-length contextual input without an explicit pooling step. We experimentally compare this Transformer-based variant with our MLP + mean-pooling design in App. B.

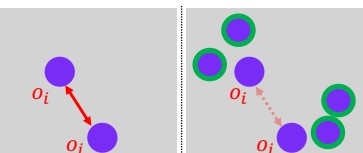

Figure 3: **Structured perception by GRMs.** Without contextual objects (left), group identification becomes ambiguous. GRMs leverage global context (right) by aggregating embeddings from all objects (green contour) into $\mathbf{o}_{ij}^*$.

The resulting affinity score $s_p$ indicates the confidence that two objects belong to the same group. GRMs then threshold these affinities to extract groups and aggregate attributes such as color diver-

sity and shape uniformity, producing a structured symbolic feature representation for each detected group.

## 3.2 SYMBOLIC ABSTRACTION

Gestalt Rule Models (GRMs) use program induction to generate interpretable, Gestalt-based rules from visual input. This is achieved by translating perceptual features into a first-order logic representation suitable for rule learning.

First, GRMs convert observations into atomic formulas (facts) using domain-specific predicates, yielding two distinct fact sets. The first, $\mathcal{F}^{\text{obj}}$, encodes object-level properties such as shape, color, and position (e.g., `shape(obj1, circle)`, `pos(obj1, 5, 10)`). The second, $\mathcal{F}^{\text{grp}}$, captures group-level structures like Gestalt principles and membership relations (e.g., `principle(g1, proximity)`, `member(obj1, g1)`).

From this symbolic representation, GRMs construct an initial candidate clause set, $\mathcal{R}_{\text{init}}$. This set contains simple clauses, each composed of a head and body atom, that serve as a starting point for a subsequent search step where they are expanded into more complex clauses. By bridging perceptual processing with logical reasoning, GRMs enable the discovery of compositional clauses that explain complex visual patterns.

## 3.3 RULE LEARNING WITH STRUCTURED PERCEPTION

GRMs leverage the initial candidate clause set $\mathcal{R}_{\text{init}}$ to learn *target rules*[2] that explain visual scenes according to Gestalt principles. For every candidate clause in $\mathcal{R}_{\text{init}}$, GRM create two variants: an existence clause and a universal clause. They are defined as below:[3]

**Existential Clauses ($\exists$)** are satisfied if **at least one** group in a scene meets a specific condition. For example, the rule for Fig. 4(a) requires that "there exists a group containing objects of the same color."

**Universal Clauses ($\forall$)** are satisfied only if **every** group in a scene meets a specific condition. For instance, the rule for Fig. 4(b) requires that "all groups must contain both a yellow and a blue object."

Learning then proceeds via a top-$k$ beam search over $\mathcal{R}_{\text{init}}$ for at most $T$ expansion steps: in each step, every clause is assigned a soft confidence score determined by its head (see App. C for the scoring equations), clauses are ranked by confidence, and only the top-$k$ clauses are expanded to uncover the scene's underlying group structures. The search terminates either after $T$ steps or early when a high-confidence rule has been found, with an overall computational complexity of $O(|C|^T \cdot t_C)$, where $|C|$ is the number of clauses considered and $t_C$ is the cost of evaluating a clause.

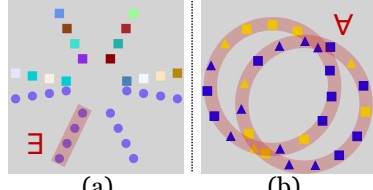

(a)  (b)

Figure 4: **GRM Scoring with quantifiers.** GRMs use existential ($\exists$) and universal ($\forall$) quantifiers to define rules over groups of objects as shown in the red shade.

## 3.4 INFERENCE

With the learned rules, GRMs finally perform inference on new inputs. At test time, the model performs interpretable inference by applying its learned symbolic rules to a given visual scene. This process unfolds in three stages. First, in the symbolic grounding stage, a perceptual backbone analyzes the input image to detect objects and groups, converting them into a logical fact base ($\mathcal{F}_{\text{test}}$). Next, during rule evaluation, each rule from the final learned set ($\mathcal{R}_{\text{final}}$) is matched against this fact base to compute a soft satisfaction score ($s_j \in [0, 1]$), quantifying its relevance under potential perceptual uncertainty. Finally, the *high-confidence prioritization strategy* determines the output: if any high-confidence rules are activated, the prediction is

---

[2]We distinguish between *clauses*, the intermediate representations manipulated during the search, and *rules*, the final explanatory solution.

[3]If no group-based patterns are found, GRMs can treat each object as an individual group. This fallback allows them to handle non-Gestalt patterns, such as detecting the presence of a single red triangle in the image.

derived exclusively from their outputs. In all other cases, the system falls back to a robust aggregation method, computing a confidence-weighted average over all partially satisfied rules. For the mathematical descriptions and more details, please check App. D. This two-tiered decision process ensures that transparent, highly reliable rules govern predictions when applicable, while maintaining robust performance in more ambiguous scenarios.

# 4 EXPERIMENTS

In this seciton, we evaluate Gestalt Reasoning Machines (GRM) on structured visual reasoning tasks that require both perceptual organization and symbolic abstraction. Our experiments are designed to assess GRM's performance, interpretability, and scalability by answering three key questions:

**(Q1)** How does GRM perform on visual reasoning benchmarks compared to leading neural and neuro-symbolic baselines?

**(Q2)** Does the perceptual grouping lead to more interpretable and well-structured symbolic rules?

**(Q3)** How robust is GRM to architectural ablations and increased task complexity?

Through comprehensive quantitative evaluations, qualitative analyses, and ablation studies, we demonstrate that GRM provides an accurate, interpretable, and scalable approach to neuro-symbolic reasoning, effectively guided by perceptual grouping principles.

## 4.1 EVALUATION PROTOCOL

**Dataset and Task.** We evaluate GRM on the ELVIS benchmark (Sha et al., 2025), a large-scale collection of visual reasoning tasks grounded in Gestalt principles. While existing datasets like CLEVR (Johnson et al., 2017) or RAVEN (Zhang et al., 2019) test object-centric and relational reasoning respectively, they do not require *group-centric* reasoning. ELVIS is specifically designed to fill this gap, with tasks that require models to first aggregate objects into meaningful higher-order entities based on Gestalt principles (e.g., proximity, similarity) before applying logic. To our knowledge, it is the only benchmark that systematically integrates these grouping principles into a neuro-symbolic pipeline, making it uniquely suited for evaluating GRM.

The extensive benchmark features over 100 distinct tasks for each Gestalt principle. Each task is a few-shot learning problem consisting of positive and negative example images. Positive examples adhere to a latent symbolic rule (e.g., "at least one group of similar objects is all red"), while negative examples subtly violate it. From a training set of 3 positive and 3 negative 224×224 RGB images, the model's objective is to infer the underlying rule and predict binary labels for a held-out test set of the same size, without direct rule supervision.

**Metrics.** For each task, GRM is trained on the provided image set to induce a set of explanatory rules. At test time, these rules are applied to the test images using our high-confidence prioritization strategy to generate a final prediction. We evaluate performance using two primary metrics: *Accuracy*, the proportion of correctly classified images, and the *F1 Score*, the harmonic mean of precision and recall. We report both scores averaged across all tasks associated with each Gestalt principle. To this end, we perform qualitative evaluations of the interpretability and runtime comparisons.

**Baselines.** We compare GRM against a range of baselines to situate its performance and highlight the benefits of its design. To simulate a few-shot learning scenario, all models are provided with just 3 positive and 3 negative examples for training or in-context learning, and evaluated on a held-out test set of the same size. Our baseline suite begins with *Human Performance*, measured via a web interface[4] to provide a robust reference point. We include a standard deep learning approach, the *Vision Transformer (ViT-Small)*, trained end-to-end on raw pixels to capture low-level patterns without structured reasoning. To assess the capabilities of state-of-the-art generalist models, we evaluate three *Large Multimodal Models*: LLaVA-1.5 (Li et al., 2024) (zero-shot), the 78B-parameter variant of InternVL3 (Chen et al., 2024), and GPT-5 (OpenAI, 2025), which are powerful on natural images but are challenged by ELVIS's abstract patterns. Finally, as a critical neuro-symbolic comparison,

---

[4]Link omitted for anonymous review.

Table 1: **Quantitative Performance on Gestalt Reasoning Tasks.** We compare our model (**GRM**) against baselines and human performance on the ELVIS benchmark. The table shows the mean and standard deviation for Accuracy and F1 Score across five Gestalt principles. **Bold** indicates the best-performing model in each column (excluding human performance). Cell colors are normalized to visualize relative scores, from high (green) to low (pink). GRM consistently achieves state-of-the-art or competitive results, demonstrating the strength of its group-based reasoning. Model shorthands are as follows: ViT-16-224 is ViT-B/16; Llava-Qwen-7B is LLaVA-OneVision-Qwen2-7B-SI.

| Met. | Model | Proximity | Similarity | Closure | Symmetry | Continuity |
|------|-------|-----------|------------|---------|----------|------------|
| Acc. | ViT-16-224 | 0.56 ±0.19 | 0.54 ±0.15 | 0.59 ±0.20 | 0.53 ±0.18 | 0.53 ±0.19 |
|      | Llava-Qwen-7B | 0.50 ±0.13 | 0.50 ±0.10 | 0.53 ±0.12 | 0.57 ±0.14 | 0.55 ±0.15 |
|      | InternVL3-78B | 0.53 ±0.18 | 0.62 ±0.23 | 0.70 ±0.20 | 0.59 ±0.17 | 0.72 ±0.21 |
|      | GPT-5 | **0.72**±0.23 | 0.71 ±0.22 | 0.66 ±0.23 | 0.52 ±0.14 | **0.81**±0.19 |
|      | NEUMANN | 0.58 ±0.15 | 0.52 ±0.08 | 0.71 ±0.18 | 0.53 ±0.09 | 0.50 ±0.03 |
|      | GRM | 0.71 ±0.17 | **0.72**±0.21 | **0.78**±0.16 | **0.64**±0.17 | 0.78 ±0.17 |
|      | Human | 0.97 ± 0.03 | 0.87 ± 0.13 | 0.92 ± 0.08 | 0.85 ± 0.15 | 0.98 ± 0.02 |
| F1 | ViT-16-224 | 0.50 ±0.29 | 0.45 ±0.30 | 0.56 ±0.28 | **0.48**±0.29 | 0.55 ±0.25 |
|      | Llava-Qwen-7B | 0.22 ±0.30 | 0.27 ±0.32 | 0.53 ±0.28 | 0.38 ±0.34 | 0.24 ±0.33 |
|      | InternVL3-78B | 0.32 ±0.33 | 0.45 ±0.39 | 0.59 ±0.34 | 0.36 ±0.35 | 0.52 ±0.41 |
|      | GPT-5 | 0.65 ±0.33 | 0.61 ±0.34 | 0.56 ±0.35 | 0.22 ±0.30 | 0.71 ±0.29 |
|      | NEUMANN | 0.27 ±0.37 | 0.13 ±0.24 | 0.59 ±0.36 | 0.30 ±0.35 | 0.33 ±0.33 |
|      | GRM | **0.65**±0.29 | **0.63**±0.33 | **0.78**±0.22 | **0.48**±0.35 | **0.78**±0.22 |
|      | Human | 0.96 ± 0.04 | 0.81 ± 0.19 | 0.90 ± 0.10 | 0.81 ± 0.19 | 0.97 ± 0.03 |

we evaluate *NEUMANN* (Shindo et al., 2024b), which learns object-level rules but lacks perceptual grouping, thereby serving as an ablation to isolate the contribution of GRM's core mechanism.

**Pretraining.** GRM's perception backbone uses pre-trained object and group detectors that are fixed during reasoning. An object detector identifies shapes, while separate models, one for each Gestalt principle, cluster objects into groups based on learned affinities. These detector outputs, along with attributes like color and position derived directly from the object masks, are converted into a symbolic fact base for the rule learner. Complete list of these predicates is provided in App. E.

**Hardware Requirements** In our experiments, we ran InterVL3-78B on 3 NVIDIA A100-SXM4-80GB, ran GPT-5 via API and ran rest of the models on a single NVIDIA A100-SXM4-80GB. The GRM, NEUMANN and ViT-16-224 models are runnable on a MacBook Pro with M2 Chip whereas the others cannot.

## 4.2 QUANTITATIVE AND QUALITATIVE EVALUATION

To answer **Q1**, we evaluate GRM on the ELVIS benchmark, spanning five Gestalt principles: *proximity*, *similarity*, *closure*, *symmetry*, and *continuity*. Each task shares a latent structural rule but varies in features like shape, color, and size, requiring both object recognition and group reasoning.

Tab. 1 compares GRM against five baselines (ViT, LLaVA, InternVL3, GPT-5, NEUMANN). GRM outperforms all neural and neuro-symbolic baselines on three principles and achieves its strongest result on *closure* (0.78), demonstrating that explicit grouping enhances symbolic generalization beyond purely object-centric models. Fig. 5 (middle) further analyzes performance by conditioning on object- and group-level properties. GRM maintains balanced accuracy (∼73%) across shape, color, size, group number, and group size, whereas GPT-5 shows imbalances (e.g., strong on shape but weaker on size and color). This consistency highlights the benefit of rule-based evaluation in avoiding confusion from perceptually similar attributes, generalizing to various abstract concepts.

We further evaluate the quality of symbolic fact extraction on 500 test tasks. Object-level properties are extracted with high reliability, including *size* (99%), *position* (95%), and *color* (90%). In contrast, group-level attributes are more challenging due to abstraction and perceptual ambiguity: *group label* reaches 71% accuracy, *object count* 92%, *group number* 76%, and *per-group count* only 44%. These results indicate that while low-level symbolic properties are robustly recovered, higher-order group-related facts remain a bottleneck for reasoning. Full results are provided in App. F.

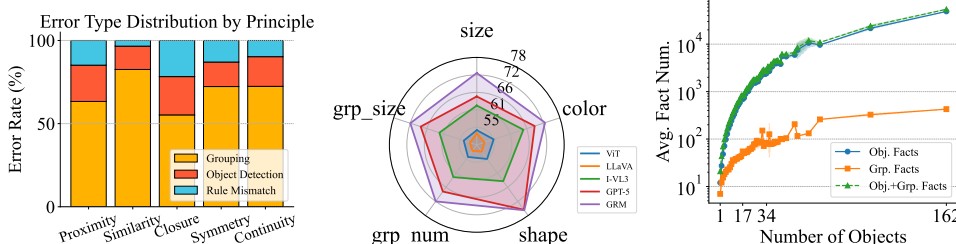

Figure 5: **Combined Results. Left**: Error breakdown by principle, showing grouping errors dominate across all Gestalt principles. **Middle**: Average accuracy (%) over all principles for each property and model. Object-level properties: size, color, shape; Group-level properties: group number and group size. **Right**: Symbolic scalability across object counts, where group-level reasoning adds moderate overhead when combined with object facts but remains lightweight when used alone.

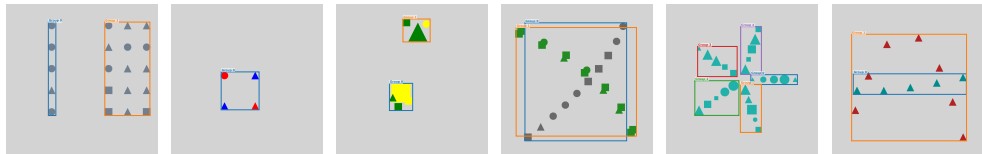

Figure 6: **Qualitative Results of Group Detection**. Visualization of predicted group structures on randomly selected tasks from the ELVIS benchmark. Each image shows the original scene with predicted group bounding boxes overlaid, demonstrating how GRM organizes visual objects according to different perceptual grouping principles. See App. F for more examples.

To answer **Q2**, we assess whether perceptual grouping improves the interpretability and structure of induced rules. Unlike black-box baselines, GRM represents visual patterns as explicit symbolic rules grounded in detected objects and groups. Fig. 6 visualizes predicted group structures (See more examples in Fig. 8 in App. F), while Listing 1 shows representative rules. Together, they reveal how GRM discovers human-readable rules (e.g., groups containing triangles or color–shape combinations) and organizes objects into coherent groupings, providing transparent reasoning traces aligned with Gestalt principles. App. G provides several task solving examples.

Listing 1: **GRM learns interpretable rules over groups.** Example rules discovered on ELVIS.

```
% Proximity principle
group_target(G,X):-has_shape(O,triangle),in_group(O,G).
    [confidence=1.000, scope=universal]
group_target(G,X):-has_color(O,red),has_shape(O,square),in_group(O,G).
    [confidence=1.000, scope=existential]
```

Fig. 5 (left) presents the distribution of GRM's error sources: grouping errors, object detection errors, and rule mismatches. Grouping emerges as the dominant source of failure, underscoring that the transition from objects to coherent groups remains the most challenging stage. A common issue is the grouping module mistakenly merging distinct structures into a single group, which disrupts the construction of correct symbolic facts. For instance, in a *closure* task (see Fig. 8 in App. F), separate closure groups are erroneously combined, preventing the reasoning module from deriving correct group-level facts and leading to incorrect predictions. Currently, our grouping mechanism uses relatively simple neural networks. Developing more robust and semantically informed grouping mechanisms is a promising avenue for future work.

Beyond grouping, object detection errors constitute the second-largest category, often caused by subtle color variations or incomplete contours due to occlusion or truncation. Rule mismatches account for the remaining cases, reflecting situations where both detection and grouping are correct but the reasoning module still fails to align symbolic predicates with the intended rules. These results highlight that while GRM successfully integrates perceptual and symbolic components, future improvements require both more robust grouping strategies and refined reasoning mechanisms to reduce systematic errors.

## 4.3 ROBUSTNESS AND COMPONENT ANALYSIS

To answer **Q3**, we analyzed GRM's robustness through ablation studies and scalability tests.

**Ablation of Perceptual Grouping.** We performed an ablation study to quantify the contribution of the grouping mechanism. As shown in Tab. 2, we compare the full GRM model (*w/ Group*), which uses both object- and group-level facts, against a variant that reasons only over object-level features (*w/o Group*), which correspond to conventional neuro-symbolic systems. The results demonstrate that incorporating group-level information consistently and significantly improves accuracy, with the most dramatic gains on tasks requiring *continuity* (+61%) and *similarity* (+36%). This confirms that group-level abstraction provides a powerful inductive bias for structured reasoning.[5]

**Scalability with Scene Complexity.** We also assessed scalability by measuring the growth of the symbolic fact base as the number of objects in a scene increases (Fig. 5, right). While a purely object-based representation grows near-quadratically, adding group-level facts introduces only a modest representational overhead. This small cost yields a substantial performance benefit, as evidenced by the accuracy gains in Tab. 2. Perceptual grouping therefore acts as a lightweight yet highly effective enhancement to the symbolic pipeline. It enriches the model's representational capacity and provides a natural mechanism for abstracting away redundant object-level details, pointing to promising directions in group-guided symbolic compression.

Table 2: **Grouping enhances the reasoning performance.** Accuracy (%) comparisons of GRM without grouping vs. with grouping. The green values are the relative improvement of the w/-Group relative to the w/o Group.

| Principle | w/o Group | w/ Group |
|---|---|---|
| Proximity | $58.0_{\pm 15.0}$ (0%) | $\mathbf{70.0}_{\pm 17.0}$ (+19%) |
| Similarity | $52.0_{\pm 8.0}$ (0%) | $\mathbf{70.0}_{\pm 22.0}$ (+36%) |
| Symmetry | $53.0_{\pm 9.0}$ (0%) | $\mathbf{62.0}_{\pm 18.0}$ (+17%) |
| Closure | $71.0_{\pm 18.0}$ (0%) | $\mathbf{79.0}_{\pm 16.0}$ (+11%) |
| Continuity | $50.0_{\pm 3.0}$ (0%) | $\mathbf{81.0}_{\pm 19.0}$ (+61%) |

**Computational Efficiency.** In terms of time efficiency, GRM is substantially more efficient than GPT-5 and remains competitive with the highly stable InternVL3-78B, solving most tasks in under 10 seconds while GPT-5 often exceeds 100 seconds (Fig. 7). A detailed analysis of how perceptual grouping quality affects induction time is provided in App. H.

Figure 7: **Task Solving Time Comparison.** The time is measured from the start of image input to the completion of rule induction.

## 5 CONCLUSION

Before concluding, let us discuss the limitations of GRM. Our experiments relied on synthetically generated visual scenes (ELVIS). While this controlled setup was essential for the first systematic study of Gestalt-based reasoning, GRM's performance on noisy, real-world images remains an open question. A natural next step is to extend GRM to more practical domains such as natural images (*e.g.*, Visual Genome (Krishna et al., 2017)). This would not only test the model's robustness but also raise important questions on how abstract Gestalt principles apply to real-world interpretation (See App. I for more discussion).

To conclude, we introduced GRM, a neuro-symbolic framework that integrates perceptual grouping with symbolic rule induction to solve complex visual reasoning tasks grounded in Gestalt principles. Our experiments demonstrate that GRM offers key advantages over purely data-driven models. The formalism of rule induction ensures logical consistency, while the scalable grouping mechanism maintains a compact symbolic representation, even in complex scenes. Together, these components enable GRM to outperform state-of-the-art models, including GPT-5, on several Gestalt reasoning principles. These results highlight the significant promise of structured neuro-symbolic architectures, positioning GRM as a foundation for developing cognitive systems with more robust, interpretable, and human-like structured perception.

---

[5]A group-only variant was not evaluated, as the reasoning tasks fundamentally require object-level predicates (e.g., shape and color), and the groups themselves are derived from detected objects.

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

## A  DETAILS ABOUT THE PERCEPTION BACKBONE

**Object Model**  The object model is a two-layer MLP: it flattens input patches, maps them to 128 hidden units with ReLU, then outputs class scores. The input patches are extracted by identifying the contours of regions in the image.

**Group Model**  The group model is trained in an end-to-end supervised manner. The dataset used for training the grouping model is synthetically generated based on the ELVIS pattern to provide ground-truth groupings based on Gestalt principles. The model takes the object embeddings as input and predicts the probability that two objects belong to the same group.

The structure of the group model includes a point encoder, a patch encoder and a classifier. A point encoder: a two-layer MLP with ReLU, mapping each input point to a hidden dimension. A patch encoder: a two-layer MLP with ReLU, mapping a flattened set of encoded points (a patch) to a patch embedding. A classifier: a two-layer MLP with ReLU, taking the concatenation of two contour embeddings and a context embedding, and outputting a single logit. The forward pass encodes two input contours and their context, concatenates their embeddings, and passes them through the classifier to produce a score.

## B  GROUP DETECTOR ARCHITECTURE COMPARISON

In the main text (Sec. 3.1), we implement GRM's grouping module using a shared encoder and principle-specific MLP heads with mean-pooled context. A natural alternative is to use a Transformer over the object set, which can in principle handle variable-length contextual input without an explicit pooling step. To validate our design choice, we performed a controlled comparison between the MLP and simple Transformer-based groupers on the ELVIS grouping task.

**Experimental Setup.**  All learned groupers are trained to solve the same binary group-detection task: given a candidate pair of objects, predict whether they belong to the same group under a given Gestalt principle. We consider three learned variants, plus a large VLM baseline:

- **MLP+Context.** The architecture described in App. A: an MLP head $h_p$ that takes $(\mathbf{o}_i, \mathbf{o}_j, \mathbf{o}_{ij}^*)$, where $\mathbf{o}_{ij}^*$ is the mean-pooled embedding of all other objects in the scene.
- **Transformer+Context.** A small Transformer encoder that takes as input the embeddings of the candidate pair and all other objects. The output corresponding to the candidate pair is pooled and fed to a classifier. This variant is meant to test whether a naïve sequence Transformer can exploit unordered, variable-size context more effectively than mean pooling.
- **Transformer Pair Only.** A Transformer that only sees the two candidate objects and ignores all other objects, thereby exploiting purely pairwise cues.
- **GPT-Zero-Shot.** A large VLM (GPT-5) prompted in a zero-shot setting to directly predict group membership for each candidate pair (no training on ELVIS).

All learned models are trained with the same data splits and loss as the main MLP grouper, and evaluated on the five Gestalt principles. Runtime is measured as average wall-clock time per ELVIS task on a single NVIDIA A100-SXM4-80GB GPU (GPT-5 timing is measured separately as an API call latency).

**Results.**  Table 3 reports per-principle accuracies, average accuracy and standard deviation across principles, runtime, and parameter counts.

**Discussion.**  Transformer-Pair-Only achieves the highest average accuracy (0.77), but it completely ignores context and mainly exploits strong pairwise cues, especially for *closure* and *continuity* where it reaches 0.97. MLP with mean-pooled context attains slightly lower average performance (0.71) but is more balanced across principles (lower standard deviation) and remains far more efficient than the GPT-zero-shot baseline.

By contrast, the naïve Transformer+Context variant, which encodes the candidate pair together with all other objects, fails to benefit from the additional context and collapses to chance level (around

Table 3: **Comparison of MLP and Transformer-based groupers.** All learned models are trained on the ELVIS grouping task; GPT-Zero-Shot denotes a large VLM (GPT-5) evaluated in a zero-shot setting. Transformer-Pair-Only attains the highest average accuracy but ignores context; MLP with mean-pooled context is slightly less accurate on average, but more balanced across principles and more efficient than GPT-zero-shot, while a naïve Transformer+Context fails to exploit context and collapses near chance.

| Metric | MLP+Context | Transformer+Context | Transformer Pair Only | GPT-Zero-Shot |
|---|---|---|---|---|
| Proximity | 0.80 | 0.50 | 0.77 | 0.82 |
| Similarity | 0.57 | 0.50 | 0.55 | 0.51 |
| Closure | 0.80 | 0.50 | 0.97 | 0.72 |
| Symmetry | 0.61 | 0.50 | 0.60 | 0.27 |
| Continuity | 0.76 | 0.50 | 0.97 | 0.82 |
| Mean Acc. | 0.71 | 0.50 | 0.77 | 0.63 |
| Acc. Std | 0.11 | 0.00 | 0.20 | 0.24 |
| Time/Task | 1.94s | 3.58s | 3.46s | 57s |
| Params | 0.5M | 3.2M | 1.6M | 635,000M |

0.50 on all principles), despite having substantially more parameters. These results suggest that the main challenge is not the mean-pooling bottleneck per se, but *how* unordered, variable-size context is encoded: a straightforward sequence Transformer over all objects does not automatically learn the relevant contextual interactions, whereas a simple MLP with permutation-invariant mean-pooled context is robust and competitive.

Designing stronger context encoders for grouping is a promising direction (e.g., more structured set-based architectures), but this is orthogonal to the main contribution of GRM, which is to show that explicit grouping combined with neuro-symbolic reasoning already yields strong and efficient performance on ELVIS.

## C    DETAILS ABOUT THE CLAUSE SCORING

During beam search, each candidate clause $r$ is evaluated by a clause scorer that estimates how well $r$ explains the task's training images while avoiding spurious matches on negatives. The score depends on the type of head attached to $r$ (`image_target`, `group_target`, or `group_universal`) and is normalized to lie in $[0, 1]$.

Intuitively, the scorer rewards coverage of positive images (or groups) and penalizes violations on negative images, thereby biasing the search towards clauses that capture stable Gestalt regularities rather than accidental coincidences.

Formally, we define:

$$\text{score}(r) = \begin{cases} \dfrac{n_+(r)}{N_+} \cdot \left(1 - \dfrac{n_-(r)}{N_-}\right), & \texttt{image\_target}, \\[2mm] \dfrac{n_+^{\exists}(r)}{N_+} \cdot \left(1 - \dfrac{n_-^{\exists}(r)}{N_-}\right), & \texttt{group\_target}, \\[2mm] \dfrac{1}{N_+}\sum_{i=1}^{N_+}\min\left(\dfrac{m_i}{M_i}, 1\right) \cdot \left(1 - \dfrac{n_-^{\forall}(r)}{N_-}\right), & \texttt{group\_universal}. \end{cases} \tag{1}$$

## D    HIGH-CONFIDENCE PRIORITIZATION STRATEGY.

Given a set of learned rules $\mathcal{R}_{\text{final}}$, each rule $r_j$ is associated with a confidence $\alpha_j \in [0, 1]$ and produces a soft match score $s_j \in [0, 1]$ on the test fact base. The final prediction score $\hat{y}_{\text{test}}$ is computed as

$$\hat{y}_{\text{test}} = \begin{cases} \frac{1}{|\mathcal{H}|}\sum_{r_j \in \mathcal{H}} s_j, & \text{if } \mathcal{H} \neq \varnothing, \\[2mm] \frac{\sum_j \alpha_j^2 \cdot s_j}{\sum_j \alpha_j^2 + \epsilon}, & \text{otherwise,} \end{cases}$$

where $\mathcal{H} = \{r_j \mid \alpha_j \geq \tau\}$ is the set of rules firing with confidence above threshold $\tau$, and $\epsilon$ is a small constant for numerical stability. A higher $\tau$ enforces stricter rule selection. It can cover more positive and fewer negative cases. which provides high precision and interpretability. For example, the threshold $\tau = 0.99$ is used to retain only those rules whose confidence exceeds $99\%$, meaning the rule is highly consistent with the training examples. In the experiments, we choose $\tau = 0.99$, which keeps only the rules that reliably distinguish positive from negative examples.

If no rule fires at all, a fixed fallback prior (e.g., $0.1$) is returned. This strategy prioritizes high-confidence rules when available, while providing a smooth weighted aggregation otherwise.

# E  SHARED BACKGROUND KNOWLEDGE FOR MODELS

For GRM, reasoning relies on a set of pre-defined predicate functions that serve as background knowledge (Tab. 4). These predicates are not learned by the model but specified in advance by the human, covering basic object- and group-level properties (e.g., shape, color, size, membership). The advantage of this design is flexibility: different tasks can be supported by simply providing different predicate sets, while the model pipeline itself remains unchanged. In this way, GRM decouples symbolic knowledge specification from the reasoning procedure, enabling interpretable and task-adaptive rule induction without modifying the architecture.

For fairness, all corresponding predicate definitions are also provided to LLM baselines in the form of natural language prompts, so that both GRM and LLMs operate with the same symbolic information. The background knowledge prompt is given as follows:

> You are given images containing multiple objects and groups. Each object and group has attributes: shape, color, size, position, and group membership. Logical patterns in the image may involve single relations (e.g., all objects have the same color) or combinations of multiple relations (e.g., objects with the same shape are grouped together and mirrored along the x-axis). You can reason about: Individual attributes: shape, color, size, position; Group properties: number of members, grouping principle; Relations: same/different shape, color, size; mirrored positions; unique/diverse attributes within groups. Analyze the image by identifying both simple and complex combinations of these relations.

Table 4: List of predicate functions used in the model.

| Predicate | Type | Description |
|---|---|---|
| has_shape | Object | Returns shape index for each object |
| has_color | Object | Returns color index for each object |
| x | Object | Returns x position for each object |
| y | Object | Returns y position for each object |
| w | Object | Returns width for each object |
| h | Object | Returns height for each object |
| in_group | Object | Returns group membership matrix |
| not_has_shape_rectangle | Object | True if object is not a rectangle |
| not_has_shape_circle | Object | True if object is not a circle |
| not_has_shape_triangle | Object | True if object is not a triangle |
| same_shape | Object | Pairwise: True if objects have same shape |
| same_color | Object | Pairwise: True if objects have same color |
| same_size | Object | Pairwise: True if objects have same size |
| mirror_x | Object | Pairwise: True if objects are mirrored along x-axis |
| same_y | Object | Pairwise: True if objects share y-coordinate |
| group_size | Group | Returns number of members in each group |
| principle | Group | Returns grouping principle index |
| no_member_rectangle | Group | True if group has no rectangle members |
| no_member_circle | Group | True if group has no circle members |
| no_member_triangle | Group | True if group has no triangle members |
| diverse_shapes | Group | True if group contains at least two shapes |
| unique_shapes | Group | True if group contains only one shape |
| diverse_colors | Group | True if group contains at least two colors |
| unique_colors | Group | True if group contains only one color |
| diverse_sizes | Group | True if group contains at least two sizes |
| unique_sizes | Group | True if group contains only one size |
| same_group_counts | Group | True if all groups have same member count |

# F  SYMBOLIC FACT EXTRACTION PERFORMANCE

Tab. 5 reports the detailed accuracy of symbolic fact extraction across 500 ELVIS test tasks. While object-level properties such as *shape* (87%), *color* (90%), *size* (99%), and *position* (95%) are recovered with high reliability, group-level properties are more challenging. *Group label* achieves 71% accuracy, *object count* 92%, *group number* 76%, and *per-group count* only 44%. These results highlight a gap between reliable low-level perception and more complex relational grouping, motivating further improvements in symbolic abstraction. Fig. 8 shows the examples of grouping results over different gestalt principles.

Table 5: Mean accuracy (%) and standard deviation of symbolic fact extraction across 500 ELVIS test tasks.

| Fact Type | Shape | Color | Size | Position | Group Label | Obj. Count | Group # | Per-Group Count |
|---|---|---|---|---|---|---|---|---|
| **Accuracy** | $0.87 \pm 0.03$ | $0.90 \pm 0.06$ | $0.99 \pm 0.01$ | $0.95 \pm 0.05$ | $0.71 \pm 0.21$ | $0.92 \pm 0.08$ | $0.76 \pm 0.24$ | $0.44 \pm 0.39$ |

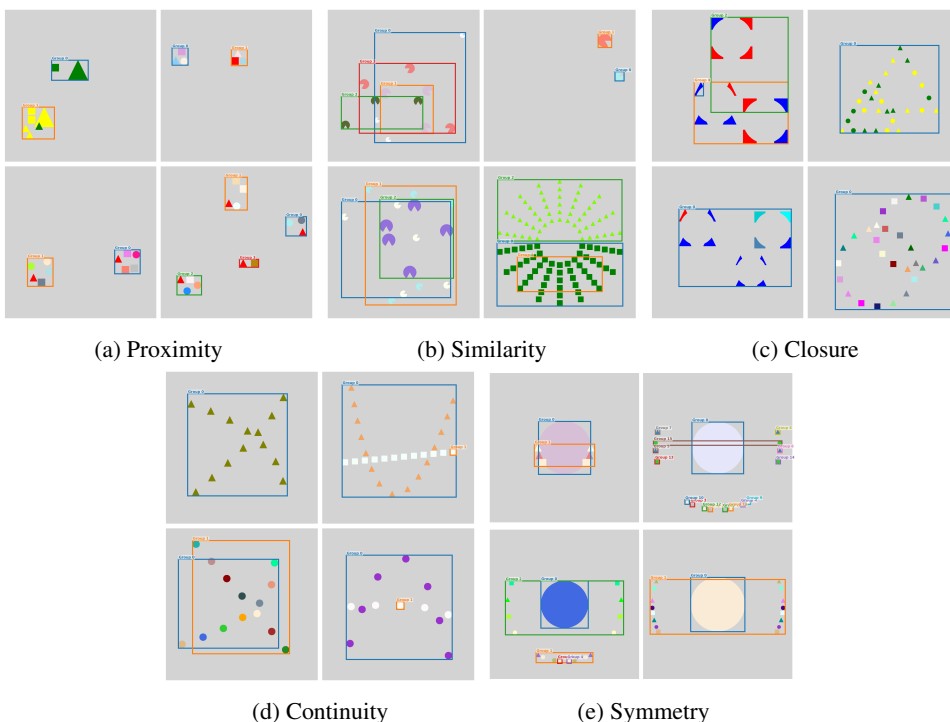

(a) Proximity  (b) Similarity  (c) Closure

(d) Continuity  (e) Symmetry

Figure 8: **Qualitative Results of Group Detection**. Visualization of predicted group structures for five Gestalt principles on randomly selected tasks from the ELVIS benchmark. Each image shows the original scene with predicted group bounding boxes overlaid, demonstrating how GRM organizes visual objects according to different perceptual grouping principles.

# G  TASK EXAMPLES AND THE MODEL ANSWERS

## G.1  EXAMPLE TASK 1

This is a task called *Triangle in Groups* following *proximity* principle. The ground-truth rule is that each proximity group contains at least one triangle. Fig. 9 presents the positive and negative examples.

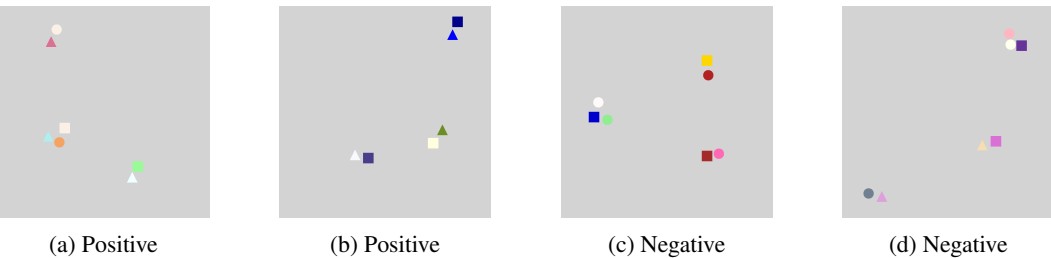

(a) Positive      (b) Positive      (c) Negative      (d) Negative

Figure 9: Triangle in Groups: Each proximity group has at least one triangle.

**GRM.** The induced rule from GRM is shown in Listing 2. The target rule is successfully identified by GRM, but its confidence is only 0.667. This relatively low value reflects imperfect object or group detection in the images (See Fig. 10 (d)), which reduces the rule's overall confidence score.

Listing 2: Rules induced by GRM on Example Task 1

```
% Group-level rules
group_target(G,X) :- has_shape(O,0), in_group(O,G).
   [confidence=0.667, scope=universal]
```

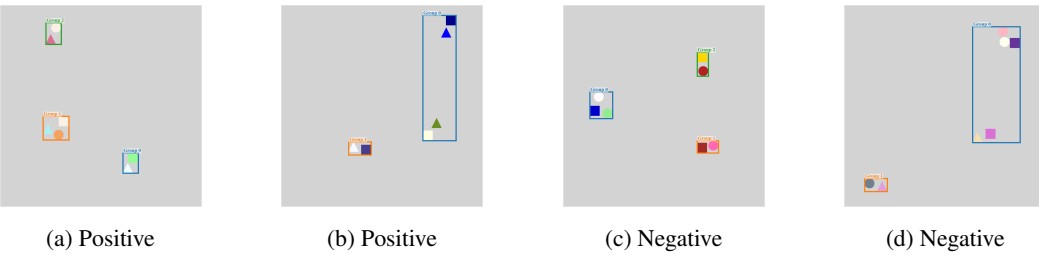

(a) Positive      (b) Positive      (c) Negative      (d) Negative

Figure 10: GRM grouping results of task 1 examples

**GPT-5** The induced logic rules by GPT-5 are shown in Listing 3. The first rule correctly identifies that each proximity group must contain a triangle, but it incorrectly constrains the group size to exactly two, whereas the ground truth allows any size of two or more. The second rule does not match the task semantics and is therefore incorrect.

Listing 3: Rules induced by GPT-5 on Example Task 1 (reformated by authors)

```
Grouping by proximity.
1. Every proximity group must be a pair of exactly two shapes:
one triangle and one non-triangle (circle or square).
2. No proximity group may contain two non-triangles or
have more/less than two members.
```

## G.2 EXAMPLE TASK 2

This is a task called *Shape of Shape* following *closure* principle. The ground-truth rule is that objects form the shape of a triangle; all the objects have the same width and height. Fig. 11 presents the positive and negative examples.

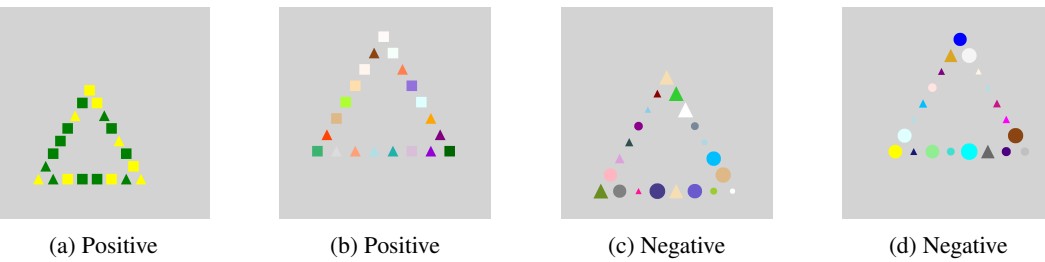

(a) Positive        (b) Positive        (c) Negative        (d) Negative

Figure 11: Big Triangle: all the objects in the image have same size.

**GRM.** The induced rule from GRM is shown in Listing 4. The target rule is successfully identified by GRM. The groups detected by the GRM are shown in Fig. 12.

Listing 4: Rules induced by GRM on Example Task 2

```
% Image-level rules
image_target(X) :- unique_sizes(I).
    [confidence=1.000, scope=image]
group_target(G,X) :- unique_sizes(G).
    [confidence=1.000, scope=image]
% Existential rule
group_target(G,X) :- unique_sizes(G).
    [confidence=1.000, scope=existential]
```

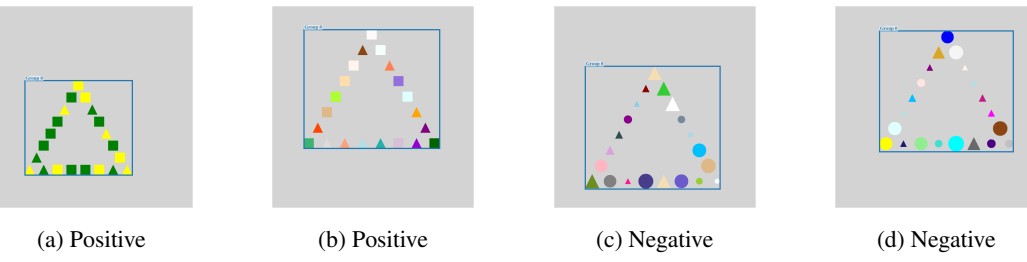

(a) Positive        (b) Positive        (c) Negative        (d) Negative

Figure 12: GRM grouping results of task 2 examples

**GPT-5.** The induced rules by GPT-5 are shown in Listing 5. Rules 1–4 are logically correct but not target rules, as they hold for both positive and negative cases. Rules 5 and 6 are incorrect and do not reflect the task semantics.

Listing 5: Rules induced by GPT-5 on Example Task 2 (reformated by authors)

```
1. Objects are positioned so that their arrangement forms an implied
closed triangular contour (three sides meeting at three vertices);
2. Small gaps are allowed, but each side is perceptually continuous;
3. Most elements lie on the triangle perimeter;
4. The interior is largely empty.
5. Colors, shapes, sizes, and groupings of the objects are irrelevant.
6. Images that do not produce a clear, closed triangular outline
(scattered points, missing sides, or filled interiors) are negative.",
```

### G.3 EXAMPLE TASK 3

This is a task called *One Splits Two* following *continuity* principle. The ground-truth rule is that objects with diverse sizes are arranged along a continuous path that later splits into two directions. Fig. 13 presents the positive and negative examples.

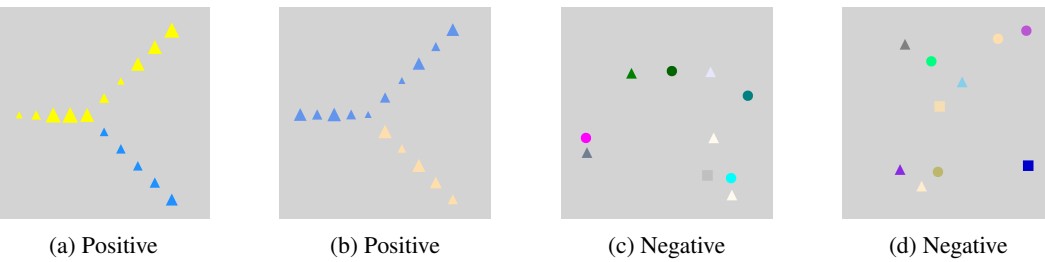

| (a) Positive | (b) Positive | (c) Negative | (d) Negative |

Figure 13: Examples of task *One Splits Two*

**GRM.** The induced rules from GRM are shown in Listing 6. The first rule focuses on the $x$-positions of two objects within a group; although unexpected, it still fits both training and test sets, illustrating that high accuracy does not always imply correctness for the expected reasons. The advantage of GRM is that such rules are interpretable and auditable: unlike black-box models, unsafe or spurious rules can be manually removed or their confidence reduced by adding targeted training examples. The second rule successfully matches the target rule, and the corresponding grouping results are illustrated in Fig. 14.

Listing 6: Rules induced by GRM on Example Task 3

```
% Image-level rules
image_target(X) :- mirror_x(O1,O2), same_color(O1,O2).
    [confidence=1.000, scope=image]
% Group-level rules
group_target(G,X) :- diverse_sizes(G).
    [confidence=1.000, scope=universal]
```

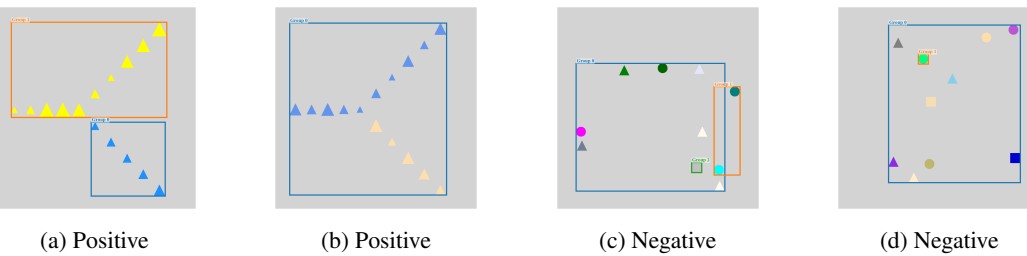

| (a) Positive | (b) Positive | (c) Negative | (d) Negative |

Figure 14: GRM grouping results of task 3 examples

**GPT-5.** The induced rules by GPT-5 are shown in Listing 7. Rules 1–4 are logically valid, with Rule 2 matching the target, but the coverage is limited: precision reaches $1.0$ while recall remains at $0.3$, showing that even when the correct rule is discovered GPT-5 does not guarantee high recall.

Listing 7: Rules induced by GPT-5 on Example Task 3 (reformated by authors)

```
1. The scene contains exactly two groups.
2. Each group is homogeneous:
all members share the same shape and the same color.
3. Members of a group are arranged along a single smooth, continuous path
(straight or gently curved), showing clear positional continuity.
4. Along each path the sizes vary gradually in one direction
(monotonic size progression).
```

# H  TASK SOLVING TIME ANALYSIS

Fig. 15 reports the average time required by GRM to induce target rules across the five Gestalt principles, with comparisons to GPT-5 and InternVL3-78B.

InternVL3-78B exhibits the most stable efficiency, requiring about 15s across all principles. GPT-5 is slower and more variable, averaging around 100s per task. On *symmetry*, GPT-5 exceeds 150s, consistent with its weaker accuracy: when uncertain, the model takes longer before committing to a prediction.

GRM is generally efficient, with rule induction on *proximity*, *closure*, and *continuity* completed in under 10s. On *symmetry*, GRM requires about 35s. The longer time is due to extended search: when no high-confidence rules are quickly available, the search process continues until a satisfactory candidate is identified *or* the maximum extended step is exceed.

The main outlier is *similarity*, where GRM averages nearly 90s. Unlike other principles, similarity depends almost entirely on color and size cues, with minimal reliance on positional features. This reduces grouping accuracy and often yields many spurious groups. A larger number of candidate groups substantially enlarges the space of possible group-level rules, thereby increasing induction time.

In summary, most GRM tasks can be solved within seconds to one minute, but tasks with weaker grouping cues or more ambiguous structures can extend to several minutes. These results highlight the impact of grouping quality on symbolic reasoning efficiency, and suggest that designing more robust symbolic features and search strategies is a promising direction for improving scalability.

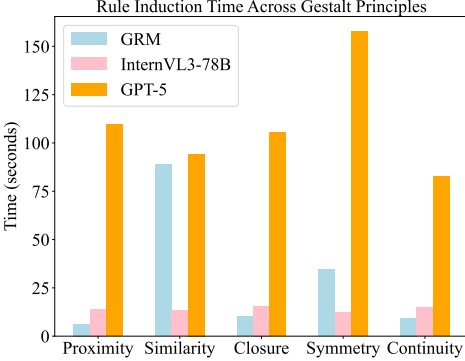

Figure 15: **Task Solving Time Comparison.** The time is measured from the start of image input to the completion of rule induction.

# I PILOT STUDY: TASK GROUPING ON COCO 2017

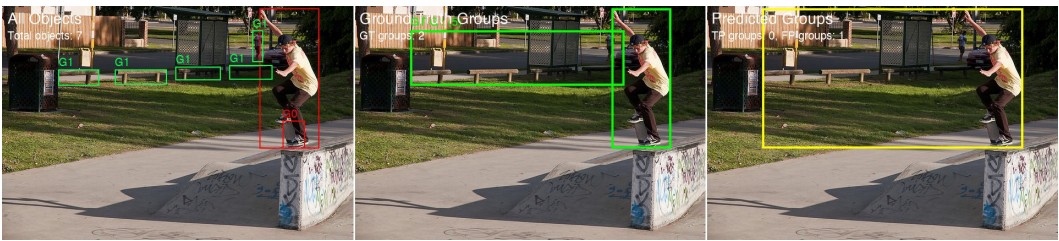

Figure 16: From left to right: labeled objects, labeled groups, predicted groups by a simple threshold-based grouper, predicted groups by a small MLP

To nevertheless probe whether GRM-style grouping can be instantiated on real images, we construct a small pilot dataset on COCO val2017. We select 30 images and manually annotate *proximity groups*: for each image, the annotator is free to mark 1–3 sets of objects that form a proximity group (e.g., a cluster of dishes on a table, a row of similar objects).

On this COCO subset, we keep the perception component fixed and only operate on the ground-truth bounding boxes. For the grouping task, we use a small *MLP-based grouper*, in the same spirit as the grouping module used in GRM, which takes as input the center position, width and height of two boxes, together with a simple representation of their neighboring objects as context, and predicts whether the pair belongs to the same group.

Qualitatively, we observe a characteristic failure mode of the proximity grouper: the model is prone to collapsing all objects in the scene into a single group. Fig. 16 illustrates a typical example. From left to right, we overlay all object boxes, the human-annotated task groups, and the MLP-based predictions. In the middle panel, the annotator groups the four bench-like seats and the nearby pedestrian as one proximity group, and the teenager and the skateboard as another. In contrast, the MLP predicts that all objects belong to the same group, and this single mega-group behavior occurs on most images in the pilot set. See Fig. 18 for more examples.

**Limitations of 2D Image Coordinates for Capturing 3D Grouping Structure**   In real-world scenes, human Gestalt grouping is grounded in 3D structure: objects share common supporting surfaces (e.g., the same ground plane or bench), occupy similar depths, or form physical assemblies (e.g., a person together with their skateboard). In contrast, datasets such as COCO only provide 2D bounding boxes on the image plane. When annotators decide which objects should form a group, they inevitably rely on their 3D understanding of the scene and on object semantics, whereas our grouping models receive only 2D geometric features (position, size, aspect ratio) and very local appearance cues. This creates an inherent mismatch between the information used to define the "ground-truth" groups and the information available to the model. In Fig.16, for example, the four seats and the pedestrian form a coherent 3D configuration on the far end of the scene; yet in

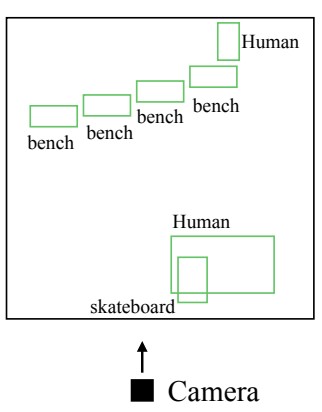

Figure 17: Top view of the Skateboard example scene in Fig. 16

the 2D projection their bounding boxes can appear similarly spaced, and perspective can bring far objects close together on the image plane. As a result, the person and his skateboard are close to the benches and the far behind pedestrian on the image. Fig.17 shows a schematic top-view of the same scene: the positions of the benches, pedestrian, skater, and skateboard clearly reveal two proximity groups, but this depth axis is absent from the original image and from its 2D annotations. To capture such structure, a grouping model would need access to explicit 3D or scene-level spatial information, such as a RGB-D image, which current 2D bounding-box datasets do not provide.

As a consequence, grouping performance on COCO-style images is difficult to interpret as a clean test of Gestalt principles: many apparent errors may reflect missing depth and scene structure rather than limitations of the grouping mechanism itself. In this work we therefore use COCO only as a

qualitative case study, and base our quantitative evaluation on synthetic stimuli where 2D geometry, grouping, and ground truth are perfectly aligned. Extending GRM to 3D object-centric representations (e.g., with estimated depth or reconstructed scenes) is an important direction for future work, and would allow us to revisit real-image grouping under conditions where the model has access to similar structural cues as human annotators.

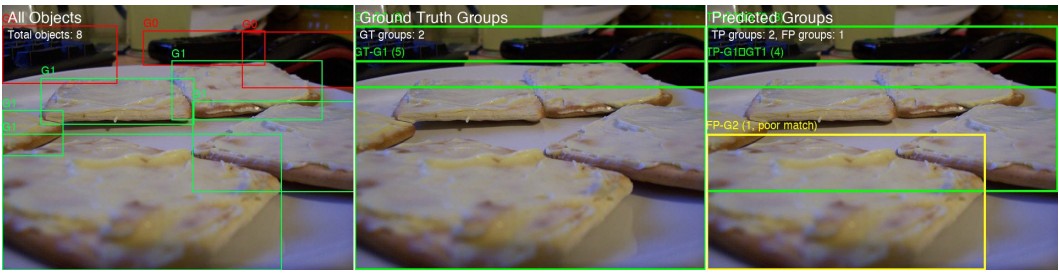

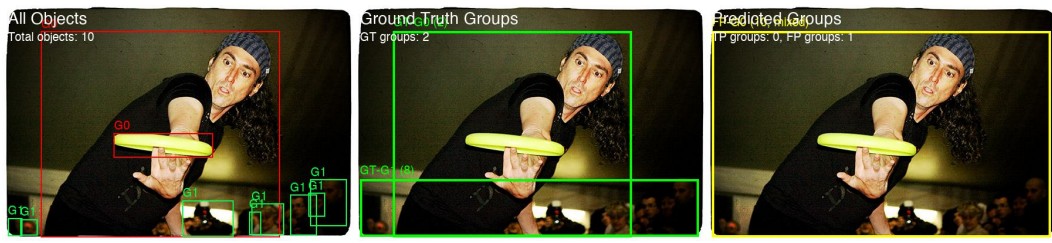

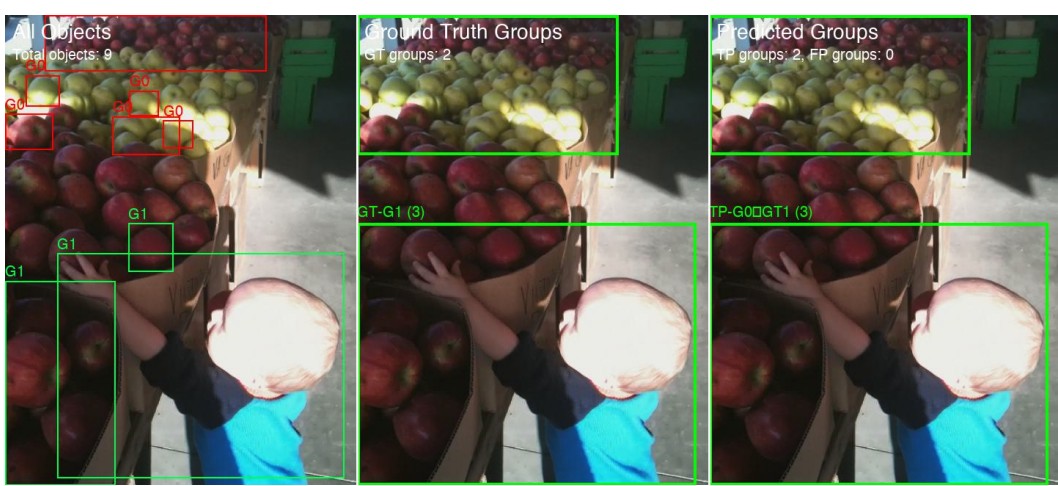

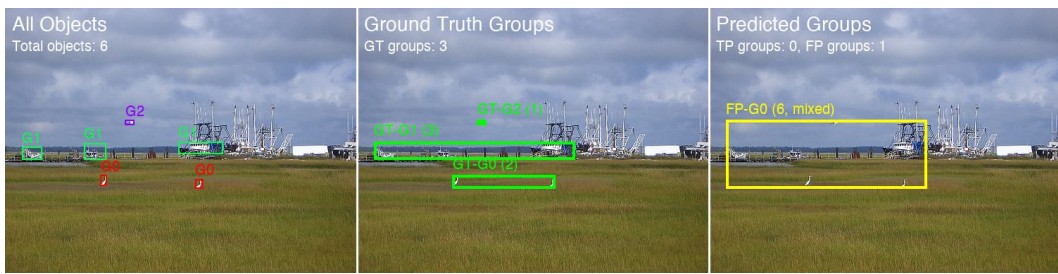

Figure 18: From left to right: labeled objects, labeled groups, predicted groups by a simple threshold-based grouper, predicted groups by a small MLP

