# OpenReview forum: "Gestalt Reasoning Machines: Structured Perception for Neuro-Symbolic Inference"
_ICLR.cc/2026/Conference — Submitted to ICLR 2026_

### Official Review · Reviewer_QuEQ · 2025-10-24

**Soundness:** 3
**Presentation:** 3
**Contribution:** 2
**Rating:** 4
**Confidence:** 4

**Summary:**

This paper advances automated visual abstract reasoning, particularly targeting Gestalt principles such as proximity, similarity, closure, symmetry, and continuity. Due to the (to-date) failure of monolithic neural approaches, this paper takes a neuro-symbolic approach where perceptual attributes are recognized with a neural network and underlying rules are recognized and applied in the symbolic domain (first-order logic). Instead of only considering object-level descriptions, this paper advances the neuro-symbolic approach by also considering group-level descriptions, which are recognized in the neural part and further processed by symbolic reasoning. Empirically, the paper shows that adding the group-level descriptions improves the reasoning capabilities on a synthetic Gestalt dataset (ELVIS), outperforming both neuro-symbolic approaches (w/o group-level) and neural approaches (small- and large-scale foundation models).

**Strengths:**

Finding the right granularity of abstraction is key in neuro-symbolic approaches. This paper demonstrates that multi-level abstractions (object- and group-level) can enhance reasoning performance. In that regard, this well written paper advances the field of neuro-symbolic learning and reasoning.

**Weaknesses:**

1.	Supervision of perception backbone. It is not clear how the perception backbone is pretrained. Are attributes values and/or groups provided as training labels? If yes, this limitation should be clearly stated and addressed with an end-to-end learning approach.
2.	Weak grouping performance. As shown in Table 4, the grouping accuracy is very low. Given that the main contribution of this paper is the grouping, it should propose and validate enhancements. First, one can question if a pure neural approach (MLP) without any inductive bias is suitable for this quite involved task. Moreover, the group-level perception (Section 3.1) accumulates all embeddings from the global context into one embedding. This averaging can certainly face some capacity limit. Having a more scalable approach that allows for concatenation (e.g., a Transformer) may improve the approach. Finally, prompting foundational models (e.g., GPT-5) to perform the grouping could be considered, too.
3.	The timing measurements are missing the neuro-symbolic baselines without group-level information (NEUMANN). Moreover, the hardware should be specified for the different methods.
4.	Finally, the evaluation is limited to only one synthetic dataset, as stated in the conclusion. While there is a pointer to another natural dataset (Visual Genome), the practical application of the proposed system is not yet justified. It would be helpful to put the work into a practical context. In which real-world applications is grouping needed?

**Questions:**

I would appreciate if the rebuttal could address the weaknesses. Besides, it would be good to specify the architectural details (neural network architecture) of GRM.

---

> ### Author Response · Authors · 2025-11-14
> **Answer for the reviewer QuEQ**
>
> Thanks for your questions and review! We answer the questions below.
>
> ### Q1. Details about perception backbone.
>
> **Object Model**
> The object model is a two-layer MLP:
> it flattens input patches, maps them to 128 hidden units with ReLU, then outputs class scores.
> The input patches are extracted by identifying the contours of regions in the image.
>
> **Group Model**
> The group model is trained in an end-to-end supervised manner.
> The dataset used for training the grouping model is synthetically generated based on the ELVIS pattern to provide ground-truth groupings based on Gestalt principles.
> The model takes the object embeddings as input and predicts the probability that two objects belong to the same group.
>
> group model structure:
> A point encoder: a two-layer MLP with ReLU, mapping each input point to a hidden dimension.
> A patch encoder: a two-layer MLP with ReLU, mapping a flattened set of encoded points (a patch) to a patch embedding.
> A classifier: a two-layer MLP with ReLU, taking the concatenation of two contour embeddings and a context embedding, and outputting a single logit.
> The forward pass encodes two input contours and their context, concatenates their embeddings, and passes them through the classifier to produce a score.
>
> ## Q2: Weak grouping performance.
>
> The grouping task is intentionally challenging, as it requires capturing complex Gestalt principles that go beyond simple feature similarity.
> The proposed grouping model shows the accuracy upto **76%** and enhance the reasoning performance over ELVIS dataset in range of **11~61%** across different tasks.
> Thus we are able to show that with a grouping model, even with imperfect accuracy, we can significantly improve reasoning performance.
>
> The main idea of this paper is not to propose a high-end grouping model, but to show that incorporating group-level reasoning can enhance visual reasoning performance.
> We design the grouping model as a simple MLP to demonstrate that even a basic neural approach can yield substantial benefits when integrated into the neuro-symbolic reasoning framework.
> We believe that **more sophisticated architectures** (e.g., graph neural networks or transformers) could further improve grouping accuracy, which we leave for future work.
>
> ## Q3: Average embedding from the global context can face some capacity limit.
>
> Thanks for your insightful suggestion.  The mean pooling is used to provide a lightweight and permutation invariant summary of the neighborhood objects,
> so the grouping module stable across different number of objects. That is the main concern we had when designing the grouping model.
>
> We agree that more expressive architectures like Transformers could capture richer global relationships and further improve grouping.
> We use simple MLPs to demonstrate the core idea.
>
> ## Q4: Prompting foundational models to perform the grouping.
>
> Thank you for the suggestion. Using advanced models such as GPT-5 can be a promising future work although the cost involved can be a bottleneck.
>
> ## Q5: timing measurements of NEUMANN.
>
> In Table 2, the time comparison only includes models that achieve reasonable accuracy such as InterVL3-78B and GPT-5.
> We would like to add the timing measurements for NEUMANN into the table. The full table is shown below:
>
> | Model              | Proximity | Similarity | Closure | Symmetry | Continuity | Average |
> |--------------------|-----------|------------|---------|----------|------------|---------|
> | GRM (Average)      | 6.28      | 88.76      | 10.29   | 34.42    | 9.32       | 29.814  |
> | GPT-5 (Average)    | 109.45    | 94.34      | 105.46  | 157.56   | 82.59      | 109.88  |
> | InterVL3-78B       | 14.11     | 13.34      | 15.58   | 12.30    | 14.89      | 14.044  |
> | NM                 | 4.34      | 48.45      | 6.80    | 24.48    | 6.34       | 18.082  |
>
> ## Q6: Moreover, the hardware should be specified for the different methods.
>
> The GRM is runnable on a personal laptop with a decent GPU (e.g., a mac book pro with M2 chip),
> whereas the large VLMs (InterVL3-78B and GPT-5) require high-end servers with multiple GPUs due to their size.
> In our experiments,  GRM ran on a NVIDIA A100-SXM4-40GB with 18% (36% peak) GPU Utilization in average
> We ran InterVL3-78B on 3 NVIDIA A100-SXM4-80GB with 40% average (100% peak) GPU Utilization
> We use GPT-5 via API, not clear about the hardware details.
>
>
> ## Q7: Grouping applications in the real-world?
> Grouping becomes essential in the multiple object reasoning scenarios.
>
> Example 1, in household or robotic planning (“clean up the after-party room”),
> a cluster of used paper cups can be considered one group and assigned to a single action in the cleaning schedule.
>
> Example 2, in off-road scenes such as a forest, individual trees do not indicate where a vehicle can safely pass.
> But when viewed as a group, trees forming a line, especially with a parallel symmetry line,
> it provides essential structural cues for navigation.
>
> ---
> **We are open to discuss further!**

---

> > ### Author Response · Authors · 2025-11-21
> >
> > Dear Reviewer,
> >
> > Please do let us know if we were able to address your concerns. If not, we will be happy to answer and discuss any remaining concerns.
> >
> > Regards,
> >
> > Authors

---

> > > ### Comment · Reviewer_QuEQ · 2025-11-24
> > >
> > > I thank the authors for their detailed response and for providing the additional timing comparisons. While the clarifications on the perception backbone (Q1) and the NEUMANN baseline (Q5) are helpful, I maintain several reservations regarding the completeness and generality of the work.
> > >
> > > Q1: Having a specific dataset for learning group models is a clear benefit for the proposed method (which other methods don’t have). This makes a fair comparison difficult.
> > >
> > > Q2: The authors state that the grouping task is "intentionally challenging" and that the simple MLP is sufficient to prove the concept. However, the lack of ablations on more capable architectures (e.g., Transformers or GNNs) is a missed opportunity.
> > >
> > > Q3: I remain concerned that the mean-pooling approach significantly bottlenecks performance. Experiments varying group sizes and embedding dimensionalities would have been crucial to prove the method scales beyond this specific synthetic setup.
> > >
> > > Q4: Since the grouping model is trained on synthetic data, a direct comparison of the grouping module's accuracy against a Foundation Model (like the "GPT-5" mentioned or GPT-4o) via prompting would have been a low-cost, high-value experiment. This would justify whether a specialized trained module is actually better than prompting a VLM for grouping.
> > >
> > > Q5: Thanks for providing the additional measurement for NEUMANN. It would be good to specify the different hardware requirements also in the table.
> > >
> > > Q6: The hypothetical examples provided (household cleaning, forest navigation) are interesting, but the paper remains limited to a single synthetic dataset. The lack of even a small-scale evaluation on real-world data (e.g., a subset of Visual Genome or real robotic scenes) makes it difficult to assess the method's practical robustness.
> > >
> > > Besides, it would be great if the paper could be adapted with the changes reported in the responses. The current version is still the original submission.

---

> ### Author Response · Authors · 2025-11-27
> **Answer of the Q1,Q2,Q3,Q4**
>
> Thank you for engaging.
>
> A1：Other methods do not have a grouping model, this is exactly the missing capability that GRM is designed to provide.
> Designing and training a grouping module for every ViT/VLM baseline is not the focus of this paper, and adding such components would fundamentally change those baselines. In fact, if we equip ViT or VLM with both a grouping model and a reasoning model “for fairness,” the resulting system would simply become another variation of GRM, rather than the original baseline. Therefore, the comparison is not unfair; it highlights that current models lack the grouping and reasoning structures that GRM explicitly introduces.
>
> A2:We additionally experimented with replacing GRM’s MLP grouping model by a simple Transformer (no architectural tricks or heavy tuning).
> We added 2 variants: Transformer+Context, which encodes the candidate pair together with all other objects, and Transformer Pair Only, which only sees the two candidate objects. The model is asked to do the group detection task only.
> Their results, together with the original MLP and the GPT-zero-shot baseline, are summarized in the table below.
> (all models (except GPT-5) are tested on a single nvidia a100-sxm4-80GB)
> | Metric           | MLP   | Transformer+Context | Transformer Pair Only | GPT-Zero-Shot |
> |------------------|-------|---------------------|-----------------------|---------------|
> | Proximity        | 0.80  | 0.50                | 0.77                  | 0.82          |
> | Similarity       | 0.57  | 0.50                | 0.55                  | 0.51          |
> | Closure          | 0.80  | 0.50                | 0.97                  | 0.72          |
> | Symmetry         | 0.61  | 0.50                | 0.60                  | 0.27          |
> | Continuity       | 0.76  | 0.50                | 0.97                  | 0.82          |
> | **Average Acc**  | 0.71  | 0.50                | 0.77                  | 0.63          |
> | **Acc Std**      | 0.11  | 0.00                | 0.20                  | 0.24          |
> | Time per Task    | 1.94s | 3.58s               | 3.46s                 | 57s           |
> | Model Parameters | 0.5M  | 3.2M                | 1.6M                  | 635000M       |
>
> Transformer-Pair-Only achieves the highest average accuracy (0.77), but it ignores context entirely and mainly exploits strong pairwise cues (especially for closure/continuity, reaches 0.97). Our GRM-MLP with mean-pooled context attains slightly lower average performance (0.71) but is more balanced across principles (lower std), while remaining far more efficient than GPT-zero-shot and much more accurate than the naïve Transformer+Context, which collapses to chance level (0.50 on all principles).
> These results suggest that the main challenge is not the mean-pooling bottleneck per se, but how unordered, variable-size context is encoded: a naïve sequence Transformer over all objects fails to exploit context, whereas a simple MLP with permutation-invariant mean-pooled context is robust and competitive. Designing a stronger encoders is a promising direction, but orthogonal to the main contribution of GRM, which is to show that explicit grouping plus neuro-symbolic reasoning already yields strong and efficient performance on ELVIS.
>
> A3: in the elvis dataset, the group numbers in a single image are in range from 1 to 4, whereas the object number is in range of 3-162, so the grouping module is trained and evaluated across more than two orders of magnitude in context size. We use mean-pooling as a permutation-invariant, size-agnostic encoder with linear cost, which makes it a pragmatic choice for GRM’s large object sets.
> Grouping itself is a semantic related tasks. We use a neural approach this time as a naive approach for that. The main goal of this paper is to propose the whole pipeline for solving the whole tasks, i.e. not only consider the grouping but also use the groups to solve the problem.
> A better grouping model, as we said, should consider the semantic meaning, which is not a trivial classification task. As we have shown. Integrating more symbolic rules into the model would be an interesting direction. And simply asking GPT-5 for everything is not a good solution. As we have shown, they are neither good at grouping or the whole task solving.
>
> A4: We ran the suggested experiment. We prompted a large VLM (“GPT-5”), for each task giving 3 example images with the same grouping logic and asking it to return a group ID for every object in the image (given the target Gestalt principle in text), and then derived pairwise same-group/different-group labels from these IDs on the same balanced test splits as our module.
> GPT-zero-shot average accuracy is 0.63, which is lower than our MLP grouping model (0.71) and is particularly unstable on some principles (e.g., 0.27 on symmetry vs. 0.61 for MLP). Moreover, GPT-zero-shot requires about 57s per task with ~6.35×10⁸ parameters, whereas our MLP uses 0.5M parameters and runs in 1.94s.

---

> ### Author Response · Authors · 2025-11-27
> **Answering Q5 and Q6**
>
> A5: The hardware requirement of NEUMANN is similar to GRM and is runnable on a personal laptop with a decent GPU.
>
> A6: We are currently running a small scale experiment for the same and will update the results as soon as we have them.
>
> We will also upload a revised version of the paper.

---

> > ### Author Response · Authors · 2025-12-01
> > **Task Grouping on COCO 2017**
> >
> > **Please Check the Appendix I in the updated version of the paper for the detailed discussion about the task grouping on COCO 2017 Dataset (in the last page)**
> >
> > To nevertheless probe whether GRM-style grouping can be instantiated on real images, we construct a small pilot dataset on COCO val2017.
> >
> > We select 30 images and manually annotate \emph{proximity groups}:
> > for each image, the annotator is free to mark 1--3 sets of objects that form a proximity group
> > (e.g., a cluster of dishes on a table, a row of similar objects).
> >
> > On this COCO subset, we keep the perception component fixed and only operate on the ground-truth bounding boxes.
> > For the grouping task, we use a small \emph{MLP-based grouper}, in the same spirit as the grouping module used in GRM, which takes as input the center position, width and height of two boxes, together with a simple representation of their neighboring objects as context, and predicts whether the pair belongs to the same group.
> >
> > Qualitatively, we observe a characteristic failure mode of the proximity grouper: the model is prone to collapsing all objects in the scene into a single group.
> >
> > ### Limitations of 2D Image Coordinates for Capturing 3D Grouping Structure
> >
> > In real-world scenes, human Gestalt grouping is grounded in 3D structure: objects share common supporting surfaces (e.g., the same ground plane or bench), occupy similar depths, or form physical assemblies (e.g., a person together with their skateboard). In contrast, datasets such as COCO only provide 2D bounding boxes on the image plane. When annotators decide which objects should form a group, they inevitably rely on their 3D understanding of the scene and on object semantics, whereas our grouping models receive only 2D geometric features (position, size, aspect ratio) and very local appearance cues. This creates an inherent mismatch between the information used to define the “ground-truth” groups and the information available to the model.
> >
> > In Fig. 16 , for example, the four seats and the pedestrian form a coherent 3D configuration on the far end of the scene; yet in the 2D projection their bounding boxes can appear similarly spaced, and perspective can bring far objects close together on the image plane. As a result, the person and his skateboard are close to the benches and the far behind pedestrian on the image. Fig. 17 shows a schematic top-view of the same scene: the positions of the benches, pedestrian, skater, and skateboard clearly reveal two proximity groups, but this depth axis is absent from the original image and from its 2D annotations. To capture such structure, a grouping model would need access to explicit 3D or scene-level spatial information, such as a RGB-D image, which current 2D bounding-box datasets do not provide.
> >
> > As a consequence, grouping performance on COCO-style images is difficult to interpret as a clean test of Gestalt principles: many apparent errors may reflect missing depth and scene structure rather than limitations of the grouping mechanism itself. In this work we therefore use COCO only as a qualitative case study, and base our quantitative evaluation on synthetic stimuli where 2D geometry, grouping, and ground truth are perfectly aligned. Extending GRM to 3D object-centric representations (e.g., with estimated depth or reconstructed scenes) is an important direction for future work, and would allow us to revisit real-image grouping under conditions where the model has access to similar structural cues as human annotators.

---

### Official Review · Reviewer_cs3A · 2025-10-29

**Soundness:** 2
**Presentation:** 3
**Contribution:** 2
**Rating:** 4
**Confidence:** 2

**Summary:**

This paper introduces GRMs (Gestalt Reasoning Machines), a neuro-symbolic framework inspired by human grouping nature.                    GRMs first use pretrained perception backbones to identify group structures, then perform rule learning based on logic search.                    This rule base can emerge at test-time and be applied to produce confidence-based inference on new images.                    Experiments on ELVIS, a synthetic dataset incorporating Gestalt principles, demonstrate the advantage of GRMs over few-shot learning neural networks or object-based neuron-symbolic methods.

**Strengths:**

1. Introducing object grouping into the visual reasoning model is novel and intuitively reasonable. This work claims as the first neuro-symbolic framework that integrates perceptual grouping with symbolic rule learning, demonstrating solid experimental results and offering great insights.
2. The overall framework is efficient, the rule induction time is much less than InternVL3-78B or GPT-5.
3. Great presentation and clear figures make it easy for the reader to follow.

**Weaknesses:**

1. Lack of detail in 4.1 ‘Pretraining’. From my point of view, the performance of GRMs is largely dependent on the pretrained perception backbones. However, the paper doesn’t include sufficient details (e.g., datasets, objectives, hyperparameters) about this.
2. Unconvincing comparison between GRMs and baselines. Based on W1, actually GRMs’ perception modules are (potentially) benefited from extensive training on images in the same distribution as the evaluation. So the experimental results in the main table cannot fully support such neural-symbolic method can outperform data-driven methods, because the evaluated VLMs may not be familiar with the test images. I would wonder whether the GRMs’ performance is still superior to VLMs after they are post-trained using the same dataset.
3. Limited generalization potential: GRMs' framework relies on predefined predicates and simplified group patterns, and cannot be used in processing real-world images.

**Questions:**

1.  Could the authors provide more reasons/evidence to support that such a neuro-symbolic method is better than large-scale data-driven training?

---

> ### Author Response · Authors · 2025-11-14
> **Answer to the reviewer cs3A**
>
> Thank you for your thoughtful comments and suggestions. We address each of your concerns below.
>
> ## Q1: Details of ‘Pretraining’.
> **Object Model** The object model is a two-layer MLP:
> it flattens input patches, maps them to 128 hidden units with ReLU, then outputs class scores.
> The input patches are extracted by identifying the contours of regions in the image.
>
> **Group Model** The group model is trained in an end-to-end supervised manner.
> The dataset used for training the grouping model is synthetically generated based on the ELVIS pattern
> to provide ground-truth groupings based on Gestalt principles.
> The model takes the object embeddings as input and predicts the
> probability that two objects belong to the same group.
> Group model structure:
> A point encoder: a two-layer MLP with ReLU, mapping each input point to a hidden dimension.
> A patch encoder: a two-layer MLP with ReLU, mapping a flattened set of encoded points (a patch) to a patch embedding.
> A classifier: a two-layer MLP with ReLU,
> taking the concatenation of two contour embeddings and a context embedding, and outputting a single logit.
> The forward pass encodes two input contours and their context,
> concatenates their embeddings, and passes them through the classifier to produce a score.
>
> ## Q2: Do GRMs still outperform VLMs after the VLMs are post-trained on the same dataset,given GRMs' perception modules may have benefited from training on that distribution?
>
> Thanks for raising this concern.
> Our goal is not to claim that GRMs universally outperform data-driven models,
> but to show that explicit group-level structure is essential for reasoning in complex multi-object scenes.
> We compare GRM with VLM to evaluate whether a purely data-driven model can perform the same type of structured,
> group-based reasoning as GRM without explicit grouping.
> For fairness,
> VLMs were also exposed to the training images with informative prompts.
> The core contribution of this work is the pipeline design: once group-level attributes are available,
> rule induction becomes feasible.
> GRM is modular, and its perception module can be replaced by a fine-tuned VLM if desired.
>
>
> ## Q3: Limited generalization potential: the framework relies on predefined predicates and simplified group patterns, and what about real-world images?
>
> GRM is intentionally designed to rely on predefined predicates. that capture general concepts,  such as color, size, position, etc.
> They are not tailored to any specific task.
> The given predefined predicates,
> they provide a set of small, universal conceptual vocabulary from which the system can compose and induce many different logical rules.
> It is not a task-specific solution.
>
> These properties (shape, color, size, etc) naturally exist in real-world images as well.
> Once object attributes are symbolized,
> (e.g. a strawberry -> red, dotted surface, rounded shape; blueberry -> blue, matt surface, rounded shape),
> the same rule induction process can be applied to real-world images.
>
> The main challenge in applying GRM to real-world scenes lies in constructing datasets that contain multi-object relational reasoning signals.
> For example, in a “clean up the after-party room” task, if multiple used paper cups are placed together somewhere in the room,
> They can be treated as one group and scheduled as a single cleaning action, rather than handling each cup individually.
> The relations among the cups will not bother the reasoning process, since they are treated as a whole thing.
> Such scenarios highlight where group-based reasoning becomes meaningful and where extending GRM to the real-world.
>
> ## Q4: Provide more reasons/evidence to support that such a neuro-symbolic method is better than large-scale data-driven training?
>
> We are not arguing that neuro-symbolic methods are universally surpass large-scale data-driven training.
> Replacing the perception component by a fine-tuned large VLM can be a promising future work.
>
> The point is that VLMs excel at recognition but lack explicit inductive bias for grouping,
> which is essential for organizing multi-object scenes into meaningful higher-level structures.
> GRM makes this structure explicit, so once the group-level attributes are available,
> the rule induction becomes tractable, interpretable and generalizes well with limited data.
>
> Empirically (see Fig. 5 middle), VLMs only match GRM on object-shape tasks.
> They struggle on object size, object color, group size, and group number reasoning, where GRM consistently performs better.
> This is because GRM compares attributes via explicit logical rules, ensuring consistent behavior.
> Finally, GRM also ranks rule confidence and selects only reliable rules (e.g. select rule with confidence 1.00 over 0.95)
> whereas VLMs often mix correct and partially-correct rules (See appendix D, GPT-5’s answer) and
> apply them inconsistently, leading to lower accuracy.
>
> ---
> **We are open to discuss further!**

---

> > ### Author Response · Authors · 2025-11-21
> >
> > Dear Reviewer,
> >
> > Please do let us know if we were able to address your concerns. If not, we will be happy to answer and discuss any remaining concerns.
> >
> > Regards,
> >
> > Authors

---

> > ### Comment · Reviewer_cs3A · 2025-11-26
> >
> > I thank the authors for their response, which has helped me better understand this work.
> > However, I still have some questions unsolved: I think the claim that ''VLMs excel at recognition but lack explicit inductive bias for grouping'' is questionable. They might not have been explicitly trained on these tasks, but once trained on this data (e.g., using the same dataset to train the group model), they can potentially do well. Willing to see some experiments on this, because if this works, it challenges the necessity of introducing GRM or explicitly group modeling.

---

> > > ### Comment · Reviewer_cs3A · 2025-11-26
> > >
> > > I also agree with the comments by Reviewer QuEQ:
> > >
> > > ''Having a specific dataset for learning group models is a clear benefit for the proposed method (which other methods don’t have). This makes a fair comparison difficult.''
> > >
> > > ''Since the grouping model is trained on synthetic data, a direct comparison of the grouping module's accuracy against a Foundation Model (like the "GPT-5" mentioned or GPT-4o) via prompting would have been a low-cost, high-value experiment. This would justify whether a specialized trained module is actually better than prompting a VLM for grouping.''

---

> > > > ### Author Response · Authors · 2025-11-28
> > > >
> > > > Thank you for engaging with the rebuttal.
> > > >
> > > > Answer to comment 1: Other methods do not have a grouping model, this is exactly the missing capability that GRM is designed to provide. Designing and training a grouping module for every ViT/VLM baseline is not the focus of this paper, and adding such components would fundamentally change those baselines. In fact, if we equip ViT or VLM with both a grouping model and a reasoning model “for fairness,” the resulting system would simply become another variation of GRM, rather than the original baseline. Therefore, the comparison is not unfair; it highlights that current models lack the grouping and reasoning structures that GRM explicitly introduces. Still, we are currently running a small scale experiment for the same and will update the results here as soon as we have them.
> > > >
> > > > Answer to comment 2: We additionally experimented with replacing GRM’s MLP grouping model by a simple Transformer (no architectural tricks or heavy tuning).
> > > > We added 2 variants: Transformer+Context, which encodes the candidate pair together with all other objects, and Transformer Pair Only, which only sees the two candidate objects. The model is asked to do the group detection task only.
> > > > Their results, together with the original MLP and the GPT-zero-shot baseline, are summarized in the table below.
> > > > (all models (except GPT-5) are tested on a single nvidia a100-sxm4-80GB)
> > > > | Metric           | MLP   | Transformer+Context | Transformer Pair Only | GPT-Zero-Shot |
> > > > |------------------|-------|---------------------|-----------------------|---------------|
> > > > | Proximity        | 0.80  | 0.50                | 0.77                  | 0.82          |
> > > > | Similarity       | 0.57  | 0.50                | 0.55                  | 0.51          |
> > > > | Closure          | 0.80  | 0.50                | 0.97                  | 0.72          |
> > > > | Symmetry         | 0.61  | 0.50                | 0.60                  | 0.27          |
> > > > | Continuity       | 0.76  | 0.50                | 0.97                  | 0.82          |
> > > > | **Average Acc**  | 0.71  | 0.50                | 0.77                  | 0.63          |
> > > > | **Acc Std**      | 0.11  | 0.00                | 0.20                  | 0.24          |
> > > > | Time per Task    | 1.94s | 3.58s               | 3.46s                 | 57s           |
> > > > | Model Parameters | 0.5M  | 3.2M                | 1.6M                  | 635000M       |
> > > >
> > > > Transformer-Pair-Only achieves the highest average accuracy (0.77), but it ignores context entirely and mainly exploits strong pairwise cues (especially for closure/continuity, reaches 0.97). Our GRM-MLP with mean-pooled context attains slightly lower average performance (0.71) but is more balanced across principles (lower std), while remaining far more efficient than GPT-zero-shot and much more accurate than the naïve Transformer+Context, which collapses to chance level (0.50 on all principles).
> > > > These results suggest that the main challenge is not the mean-pooling bottleneck per se, but how unordered, variable-size context is encoded: a naïve sequence Transformer over all objects fails to exploit context, whereas a simple MLP with permutation-invariant mean-pooled context is robust and competitive. Designing a stronger encoders is a promising direction, but orthogonal to the main contribution of GRM, which is to show that explicit grouping plus neuro-symbolic reasoning already yields strong and efficient performance on ELVIS.
> > > >
> > > > We hope we have answered all your concerns and you can reconsider your score.

---

### Official Review · Reviewer_J3Xd · 2025-10-30

**Soundness:** 3
**Presentation:** 3
**Contribution:** 3
**Rating:** 6
**Confidence:** 4

**Summary:**

This work proposes a Gestalt Reasoning Machine (GRM) that fuses human-like perceptual grouping with symbolic rule learning. It operationalizes Gestalt principles within a neuro-symbolic ILP pipeline. Experiments on the ELVIS benchmark show clear accuracy gains over neural and prior neuro-symbolic systems with especially strong results under increasing visual complexity.

**Strengths:**

- The integration of Gestalt grouping within a neuro-symbolic ILP framework is novel and well-motivated by cognitive theory. The architecture bridges perception and reasoning in a way that is both interpretable and scalable.

- The explicit use of contextual embeddings in s_p(o_i,o_j,I) = \sigma(h_p(o_i,o_j,o^*_{ij})) shows the adventage of this grouping.

- The comparison against GPT-5 and InternVL3 convincingly shows that structured reasoning can outperform massive data-driven systems.

- Equation in section Appendix A is not clear: how  τ = 0.99 was chosen, for instance?

**Weaknesses:**

- Evaluation remains restricted to synthetic Gestalt scenes.

- The contextual affinity s_p(o_i,o_j,I) = \sigma(h_p(o_i,o_j,o^*_{ij})) seems to be introduced without justification. What is the theoritical grounding? Any ablation on o^*_{ij} mean embedding?

- Regarding the rule search procedure, what are the convergence guarantees and computational complexity ?

- How sensitive the proposed freamwork to the choice of grouping thresholds s_p?

- Equation in section Appendix A is not clear: how  τ = 0.99 was chosen, for instance?

**Questions:**

see Weaknesses

---

> ### Author Response · Authors · 2025-11-17
> **Answer to the reviewer J3Xd**
>
> Thank you for your thoughtful comments and suggestions. We address each of your concerns below.
>
> ## Q1: Evaluation remains restricted to synthetic Gestalt scenes.
> The synthetic dataset is used so that we can have ground-truth group labels.
> However, GRM itself is not limited to synthetic data.
> It can be directly applied to natural scenes. However, we still lack the work to create such a dataset in the real-world.
> One challenge would be how to label the objects to the same group, since this can be very subjective.
>
> ## Q2: Theoretical grounding of grouping-pair equations
>
> The contextual affinity function estimates the confidence that two objects belong to the same group.
> This decision is inherently context-dependent, because whether two objects are “close” or “coherent”
> is defined relative to the distribution of other objects in the scene.
> This aligns with classical results on locality in relational structure,
> the interaction between a pair often depends on features in their local Gaifman neighborhood \citep{Gaifman1982}.
>
> To make the input representation permutation- and size-invariant,
> we use a mean-pooled embedding over neighboring objects to summarize this local context.
> This provides a lightweight global view sufficient for Gestalt-style relational decisions.
> We agree that more expressive context encoders (e.g., attention-based pooling or Transformers) could further improve grouping,
> and we consider this a promising direction for future work.
>
> [Gaifman 1982] Gaifman, H. 1982. On local and non-local properties. In Studies in Logic and the Foundations of Math-ematics.
>
> ## Q3: Rule search convergence guarantees and computational complexity
>
> The rule learner performs a bounded top-k beam search over the clause space,
> where candidate clauses are iteratively expanded for at most T rounds.
> Here, k denotes the number of highest-scoring clauses retained at each step,
> and T represents the maximum search depth (i.e., the number of expansion rounds).
> Since both k and T are fixed, the search converges after at most T iterations.
> The search can stop early when a high confident rule has been found.
> The overall computational complexity is
> $ O( |C|^T  \cdot  t_C) $,  where $ |C| $  is the number of candidate clauses
> and t_C is the cost of evaluating a clause, which scales linearly with the number of examples.
> This bounded search ensures tractable computation and stable convergence to high-confidence rule sets in practice.
>
> In practice, $|C|$ depends only on the concepts actually present in the scenes,
> rather than on the full vocabulary of the model’s symbolic language.
> As shown in the figure 7, the GRM is not very time-consuming in general,
> note that the object number in the task is in range up to hundreds.
>
> ## Q4: How sensitive is the proposed framework to the choice of grouping thresholds s_p?
>
> As shown in the table below, we test the F1 score of the grouping results under different grouping confidence thresholds.
> Proximity shows moderate threshold sensitivity.
> For other principles, F1 remain flat or decreasing, indicating that TP/FN/FP are not well-separated in confidence.  We chose not to fine-tune s_p per principle to avoid overfitting and keep the evaluation protocol simple and comparable across settings.
>
>
>
>
>
> | Threshold | Proximity      | Similarity     | Closure        | Symmetry       | Continuity     |
> |-----------|----------------|----------------|----------------|----------------|----------------|
> | t = 0.1   | 0.56 ± 0.26    | 0.65 ± 0.35    | 0.82 ± 0.18    | 0.75 ± 0.20    | 0.07 ± 0.15    |
> | t = 0.3   | 0.62 ± 0.23    | 0.65 ± 0.35    | 0.82 ± 0.18    | 0.75 ± 0.20    | 0.09 ± 0.15    |
> | t = 0.5   | 0.67 ± 0.22    | 0.63 ± 0.35    | 0.81 ± 0.19    | 0.63 ± 0.31    | 0.11 ± 0.15    |
> | t = 0.7   | 0.70 ± 0.21    | 0.62 ± 0.35    | 0.79 ± 0.20    | 0.48 ± 0.36    | 0.11 ± 0.13    |
> | t = 0.9   | 0.74 ± 0.21    | 0.61 ± 0.34    | 0.74 ± 0.23    | 0.41 ± 0.36    | 0.09 ± 0.12    |
>
> (p.s Continuity is the only principle with low F1, shows the potential hardness of this principle. In the GRM, we only encode object contour, color and size into the input tensor and used for the group detection.
> Considering to include more global cue such as tangent/orientation information or other more effective feature representation can be a future work.)
>
>
>
> ## Q5: Equation in section Appendix A is not clear: how τ = 0.99 was chosen, for instance?
> The threshold τ = 0.99 is used to retain only those rules whose confidence exceeds 99%,
> meaning the rule is highly consistent with the training examples. In practice,
> this keeps only the rules that reliably distinguish positive from negative examples.
>
> A higher τ enforces stricter rule selection. It can cover more positive and fewer negative cases. which provides high precision and interpretability.
>
> ---
> **We are open to further discuss!**

---

> > ### Author Response · Authors · 2025-11-21
> >
> > Dear Reviewer,
> >
> > Please do let us know if we were able to address your concerns. If not, we will be happy to answer and discuss any remaining concerns.
> >
> > Regards,
> >
> > Authors

---

### Official Review · Reviewer_xYyP · 2025-10-31

**Soundness:** 3
**Presentation:** 2
**Contribution:** 3
**Rating:** 6
**Confidence:** 3

**Summary:**

Compared to traditional models relied on large datasets, this paper introduces Gestalt Reasoning Machines (GRMs), which integrates Gestalt principles to enhance reasoning models with perception capabilities. This paper demonstrates that GRMs outperform purely neural baselines in visual reasoning tasks that need perception of higher-order structures. GRM processes an image through perceptual detection, symbolic abstraction, and rule learning to derive interpretable logical rules, which are then applied by an inference engine that prioritizes high-confidence, transparent reasoning for visual prediction.

**Strengths:**

1.	Novelty: This work systematically integrate Gestalt perceptual principles (proximity, similarity, closure, symmetry, continuity) into a neuro-symbolic reasoning framework, bridging low-level perception and symbolic reasoning.
2.	Empirical Validation: Comprehensive experiments including human test on the newly proposed ELVIS benchmark demonstrate clear advantages over both neural (e.g., ViT) and large multimodal models (e.g., GPT-5, InternVL3).
3.	Efficiency: GRM achieves strong accuracy with significantly lower rule induction time.

**Weaknesses:**

1.	Data Limitation: All experiments are conducted on the ELVIS dataset.The paper mentioned that “To our knowledge, it is the only benchmark that systematically integrates these grouping principles into a neuro-symbolic pipeline, making it uniquely suited for evaluating GRM.” While this choice is reasonable for testing Gestalt-based grouping, it also raises concerns about generalization. Since ELVIS is specifically designed around grouping-centric reasoning, GRM’s advantage may partially stem from the dataset’s alignment with its inductive bias. To fully assess its robustness and versatility, it would be valuable to evaluate GRM on visual reasoning benchmarks that do not explicitly require grouping, such as CLEVR or RAVEN, to determine whether the proposed mechanism still provides benefits in more conventional reasoning settings.
2.	Writing and Presentation: Page 8 appears quite dense. The discussion on future work in the figure 5 “Currently, our grouping mechanism uses relatively simple neural networks. Developing more robust and semantically informed grouping mechanisms is a promising avenue for future work”could be moved to the conclusion section. Doing so would improve the overall flow and logical structure of the paper.

**Questions:**

1.	Since ELVIS is specifically designed around Gestalt grouping principles, have you tested (or do you plan to test) GRM on visual reasoning datasets that do not require explicit grouping, such as CLEVR or RAVEN?
2.	I am a bit confused about Table 2 and Figure 7. Could the authors clarify how the accuracy improvements reported in Table 2 were calculated? Similarly, how was the time reported in Figure 7 measured?

---

> ### Author Response · Authors · 2025-11-15
> **Answer to the reviewer xYyP**
>
> Thanks for your questions and review! We answer the questions below.
>
>
> ## Q1:  Why do not show the results on CLEVR and RAVEN?
>
> We did consider these two dataset when we designed the experiment part.
> Here are the reasons that we didn’t use them in the end.
>
> Applying GRM to CLEVR does not introduce new challenges because
> CLEVR tasks focus on object-centric relational reasoning (e.g., left-of, behind, same-color).
> This can be considered as a special case of ELVIS dataset where each scene contains only one group.  GRM is able to reasoning rules using object-level facts only, but reduces GRM to the NEUMANN-style setting.
> Indeed, ELVIS already includes comparable scenarios (e.g., scenes with only one group),
> where object-level reasoning suffices.
>
> RAVEN targets a different reasoning paradigm:
> analogical matrix reasoning and rule completion based on transformations such as rotation, mirroring, etc.
> These tasks do not involve perceptual grouping or group-level properties,
> which are the core focus of GRM.
> A key feature of GRM is that it can capture relations that only emerge at the group level,
> such as group size, group uniformity, or higher-order Gestalt patterns. These are not present in RAVEN.
> Thus, RAVEN does not align with the problem setting addressed in this work.
>
>
> ## Q2: Clarify how the accuracy improvements reported in Table 2 were calculated
>
> The accuracy improvement percentages (shown in green in Tab. 2) are computed relative to the baseline model.
> It indicates the performance gain achieved by incorporating group-level reasoning.
> The formula used is as follows:
>
> $$improvement\\% = \frac{acc_{GRM} - acc_{base}}{acc_{base}} \times 100\\%$$
>
>
> ## Q3: How was the time reported in Figure 7 measured?
> The reported runtime is measured from the start of object detection to the completion of rule induction,
> thereby covering the entire pipeline (perception + reasoning).
>
> For vision–language models (VLMs),
> the runtime is measured from the given of the prompt input to the generation of the final rule output,
> so it ensures consistency across model comparisons.
>
> ---
> **We are open to discuss further!**

---

> > ### Author Response · Authors · 2025-11-21
> >
> > Dear Reviewer,
> >
> > Please do let us know if we were able to address your concerns. If not, we will be happy to answer and discuss any remaining concerns.
> >
> > Regards,
> >
> > Authors

---

### Meta-Review · Area_Chair_WSd4 · 2026-01-05

**Summary:**

While the paper presents a technically functional neuro-symbolic framework (Gestalt Reasoning Machines), ultimately the contribution is limited by its experimental design and scope. The recommendation to reject stems from two fundamental issues regarding the fairness of comparison and the significance of the findings.

1. Unfair comparison and restricted methodology: As noted by Reviewer QuEQ and echoed by cs3A, the proposed method enjoys a distinct advantage: it utilizes a dedicated grouping module trained on a specific synthetic dataset (ELVIS) designed to provide ground-truth grouping labels. Standard baselines (like ViTs or VLMs) are not afforded this explicit, task-specific supervision. Consequently, the comparison is essentially between a system engineered with the exact solution structure (and trained on the answer key) against general-purpose models. This does not demonstrate that the neuro-symbolic approach is inherently superior, but rather that providing task-specific supervision improves performance on that specific task—a result that is expected rather than surprising.

2. Lack of generalization and emergence: The paper introduces specific human priors (Gestalt principles) to solve synthetic tasks that were explicitly generated to require those priors. This creates a circular validation loop. The work shows that if you bake in the specific rules required to solve a synthetic puzzle, the model solves the puzzle. It lacks the "surprising generalization" or "emergence of abilities" that characterizes high-impact representation learning research. The method is highly tailored to the synthetic ELVIS domain; as the authors admitted in the rebuttal (regarding COCO), applying this to real-world images is fraught with challenges because 2D bounding boxes do not capture the 3D semantic grouping humans rely on. Without demonstrating that these priors help in diverse, out-of-distribution, or real-world scenarios (as noted by Reviewer xYyP regarding CLEVR/RAVEN), the work remains a narrow solution to a constructed problem.

**Reviewer Concerns:**

The major concerns from xYyP, QuEQ and cs3A are still outstanding, i.e., slightly unfair comparison and limited scope.

**Reviewer Scores:**

The first two reviewers gave positive scores while did not participate the discussion. I don't think they would increase the scores though. The last two reviewers gave negative scores and participated the discussion (acknowledged better understanding but also raising more concerns). They are unlikely to increase the scores.

---

### Decision · Program_Chairs · 2026-01-26

Reject